# Albedo changes caused by future urbanization contribute to global warming

Zutao Ouyang [1,2 ✉], Pietro Sciusco [2], Tong Jiao[3], Sarah Feron[1,4,5], Cheyenne Lei[2], Fei Li[6], Ranjeet John [7], Peilei Fan [8], Xia Li [9], Christopher A. Williams [3], Guangzhao Chen [10,11], Chenghao Wang [1] & Jiquan Chen [2 ✉]

The replacement of natural lands with urban structures has multiple environmental consequences, yet little is known about the magnitude and extent of albedo-induced warming contributions from urbanization at the global scale in the past and future. Here, we apply an empirical approach to quantify the climate effects of past urbanization and future urbanization projected under different shared socioeconomic pathways (SSPs). We find an albedo-induced warming effect of urbanization for both the past and the projected futures under three illustrative scenarios. The albedo decease from urbanization in 2018 relative to 2001 has yielded a 100-year average annual global warming of 0.00014 [0.00008, 0.00021] °C. Without proper mitigation, future urbanization in 2050 relative to 2018 and that in 2100 relative to 2018 under the intermediate emission scenario (SSP2-4.5) would yield a 100-year average warming effect of 0.00107 [0.00057,0.00179] °C and 0.00152 [0.00078,0.00259] °C, respectively, through altering the Earth's albedo.

[1] Department of Earth System Science, Stanford University, Stanford, CA 94305, USA. [2] Center for Global Change and Earth Observations and Department of Geography, Environment & Spatial Sciences, Michigan State University, East Lansing, MI 48823, USA. [3] Graduate School of Geography, Clark University, Worcester, MA 01610, USA. [4] Campus Fryslan, University of Groningen, Wirdumerdijk 34, 8911 CE Leeuwarden, The Netherlands. [5] Department of Physics, University of Santiago, Ave Bernardo OHiggins 3363, Santiago de Chile, Chile. [6] Grassland Research Institute, Chinese Academy of Agricultural Sciences, Hohhot 010010, China. [7] Department of Biology, and Department of Sustainability, University of South Dakota, Vermillion, SD 57069, USA. [8] Center for Global Change and Earth Observations and School of Planning, Design, and Construction, Michigan State University, East Lansing, MI 48823, USA. [9] Key Lab of Geographic Information Science, School of Geographic Sciences, East China Normal University, Shanghai 200241, China. [10] Division of Landscape Architecture, Faculty of Architecture, The University of Hong Kong, Hong Kong 999077, China. [11] Institute of Future Cities, The Chinese University of Hong Kong, Hong Kong 999077, China. ✉email: ouyangzt@stanford.edu; jqchen@msu.edu

Urbanization-induced land use and land cover change (LULCC) is one of the most phenomenal changes across the terrestrial surface of the Earth, especially in recent decades. According to the MODIS land use land cover product, the global urban land was 0.79 million km$^2$ in 2018, and by 2050, 68% of the world population is expected to reside in urban areas[1], resulting in a forecasted urban land of 1.04–1.90 million km$^2$ [2–4]. These shifts will fundamentally change physical properties of the land surface, which in turn will alter the surface energy balance, hydrological cycle, and biogeochemical processes, inducing a series of environmental and climate consequences[5–8].

Anthropogenic LULCC, including urbanization defined as the expansion of urban land here, affects local-to-global climate not only through modifications in surface roughness, carbon processes, and turbulent heat fluxes, but also through changes in surface albedo that directly alter the Earth's radiation budget[9]. However, most of the attention thus far has been focused on quantifying changes in carbon processes[10–14], with less known about albedo changes caused by LULCC and the associated climate impact[9,15–18]. Collectively, for all forms of LULCC, Ghimire et al. reported that global albedo changes during 1700–2005 were responsible for a negative radiative forcing (RF) of −0.15 ± 0.10 W/m$^2$[17]. The Intergovernmental Panel on Climate Change (IPCC), on the other hand, estimated a RF of −0.2 ± 0.2 W/m$^2$ for albedo changes caused by LULCC relative to 1750, but with medium-low confidence, suggesting a need for more robust research in this area[9]. The insufficient understanding of the albedo-induced climate effect due to LULCC has recently triggered a lot of interest in climate forcing of albedo changes due to deforestation/afforestation, because forests cover a large portion of land surfaces and are promoted as a leading nature-based solution for climate change mitigation[6,19–21]. Unfortunately, very limited efforts have been made to study the climate effects of albedo change induced by urbanization, one of the most important LULCC processes that is mainly caused by human activities.

Past work quantifying the albedo consequences of global LULCC mostly used very coarse-gridded climate models (i.e., coarser than 2° spatial resolution), where the unique contribution from urbanization was not discernible because the small proportion of urban area was either assumed to be a constant through time or not properly represented[18,21,22]. We argue that the aggregated impacts of urbanization at regional-to-global scales are not negligible even though urban lands occupy only a very small proportion of global land surfaces[5,7].

Urbanization-induced albedo changes or manipulating the albedo of urban land have been shown to affect local, regional, and even global climate. Hu et al.[23] assessed the surface albedo change in Beijing from 2001 to 2009 caused by urbanization using remote sensing and field measurements. They found a positive relationship between albedo-induced RF and urbanization level and suggested that the cumulative effects of albedo change caused by urbanization could be important drivers of recent local climate change. Zhao et al.[24] found empirical evidence that the night-time urban heat island (UHI) intensity and the urban-rural albedo difference are negatively correlated and argued that increasing urban albedo can produce measurable results to mitigate UHI on a large continental scale. There also exist many simulation studies that focus on assessing the global climate effect of modifying urban albedo (i.e., with prescribed albedo changes)[25–29]. These studies, consistently show cooling effects by increasing urban albedo. However, at the global scale, no study has assessed the climate impact of surface albedo changes due to actual or projected urbanization, i.e., the change of albedo due to replacing other lands with urban lands. Overall, the albedo of urban lands is 0.01–0.02 lower than that of adjacent croplands[30], therefore urbanization can potentially have a global warming effect by reducing the Earth's albedo.

In this study, we aim to understand albedo-induced climate effects for the past (2018–2001) and the projected future (2050 relative to 2018 and 2100 relative to 2018) global urbanization by estimating its RF. We adopted a method used by Ghimire et al.[17], which integrates the spatially and temporally explicit albedo product of MODIS[31] and the LULCC databases and has yielded results comparable to those of process-based modeling work as summarized in the IPCC's Fifth Assessment Report[9]. Using a similar approach, we estimated the albedo-induced RF of historical urbanization with MODIS land cover product, and of future urbanization with a recently released future urban land projection at 1-km resolution[3] under selected shared socioeconomic pathways (SSPs). The albedo-induced RF was then converted to a carbon dioxide equivalent ($CO_{2\text{-eq}}$) metric called global warming potential (GWP), and to global mean temperature response to facilitate interpretation for decision-makers and assure simple comparisons with greenhouse gas effects.

## Results and discussion

**Surface albedo change.** Large spatial variations in surface albedo changes exist among urban landscapes (grid-level) for historical and future urbanization, which are due to a combination of local climate conditions and land cover composition changes (Fig. 1, Supplementary Figs. 1, 2). The changes in grid-level albedo fall generally between −0.001 and 0.001, because urban areas account for a very small portion of most grids and of the global land surfaces (<3% even for the projected more urbanized future in 2100). However, relatively large changes that reach <−0.005 also appear in the near (2050) and far (2100) futures relative to 2018. The areas of declined albedo are mostly in regions that used to and still have a high percentage of croplands and urban lands, including Europe, the Middle East, the east and west coasts of the United States, northern India, the North China Plain, the China Bohai Rim region, and the east coastal area of the Mediterranean Sea. More importantly, the spatial pattern of grids with negative changes in albedo is similar to the global distribution of croplands (Supplementary Fig. 3), suggesting that most urban lands continue to be converted from adjacent croplands[2,32,33] that, on average, have higher surface albedo than urban lands (Supplementary Fig. 4). The frequency distribution of global grid-level albedo is left skewed (Fig. 1d), leading to a global mean of reduced albedo that produces positive RF (see Radiative Forcing).

The land cover change that caused the observed albedo change at grid-level, is dominated by a replacement of croplands, savannas, and grasslands with urban lands. The land cover that loses most of the area at grid-level is dominated by croplands, savannas, and grassland in both the past and the projected futures, especially in the temperate zone, but there also is a major loss of forests in some higher northern areas and losses of both forests and water in Amazon regions (Fig. 2). This spatial pattern is consistent among different future scenarios (Supplementary Fig. 2).

**Radiative forcing.** Urbanization worldwide produces a warming effect through reducing albedo in both the past and the projected futures. Relative to 2001, the increased urban lands in 2018 produced an RF of ~0.00017 W/m$^2$ (~0.18 Gt $CO_{2\text{-eq}}$) through changes in albedo, which led to a warming of the 100-year mean global surface temperature of ~0.00014 [0.00008, 0.00021] °C. For the future, the RF due to albedo changes caused by urbanization relative to 2018 is estimated to be positive and constantly increasing under three illustrative scenarios, albeit the increasing rate will slow down after 2050 for SSP1-2.6 and after 2070 for SSP2-4.5 (Fig. 3). These increasing trends suggest that urban

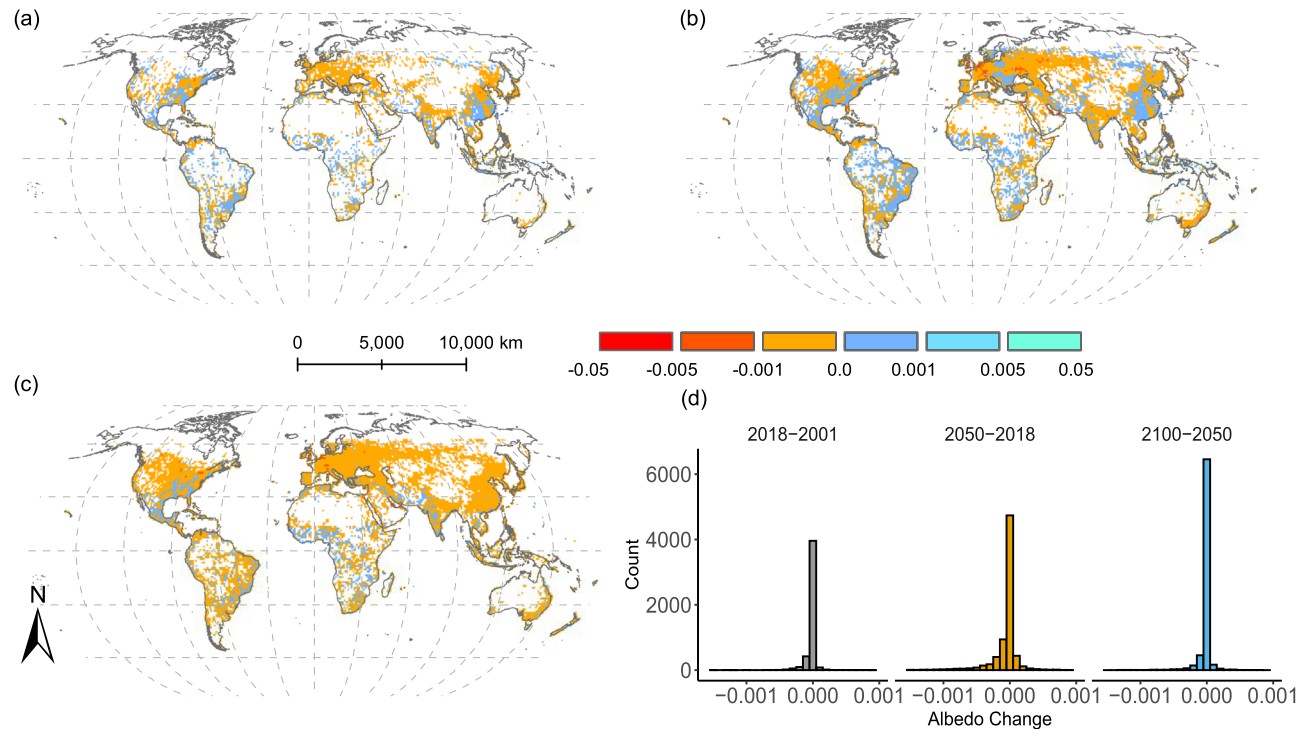

**Fig. 1 Global changes in land surface albedo due to urbanization. a** Changes in 2018 relative to 2001, **b** changes in 2050 relative to 2018, **c** changes in 2100 relative to 2050, and **d** the histogram showing the full distribution of the grid-level albedo changes. **b**, **c** are albedo changes under the SSP2-4.5 scenario. The albedo changes under SSP1-2.6 and SSP5-8.5 scenarios are provided in Supplementary Fig. 1 and show similar patterns as under SSP2-4.5.

expansions are projected to keep replacing lands that on average have higher albedo values regardless of SSP scenarios. It is worth noting that snow cover can play an important role in estimating global RF from future urban expansions under different scenarios, as surface albedo can increase significantly when land is covered by snow (Supplementary Fig. 4). This albedo-induced warming effect (i.e., positive RF) from urban expansion is additional to the warming effect of reduced carbon sequestration capability that is induced by replacing fertile lands with construction materials[14,34,35].

Future urbanization imposes an even greater warming effect under higher emission scenarios (i.e., SSP5-8.5 vs. SSP2-4.5 vs. SSP1-2.6), which project more urban expansion and a higher temperature increase. The estimated global RF is 0.00128 [0.00102,0.00206] W/m$^2$ (~1.37 [1.09, 2.20] Gt $CO_2$-eq) under SSP1-2.6, and it increases to 0.00133 [0.00107, 0.00202] W/m$^2$ (~1.42 [1.15,2.16] Gt CO2-eq) under SSP2-4.5, and further increases to 0.00158 [0.00123, 0.00240] W/m$^2$ (~1.69 [1.32, 2.57] Gt $CO_2$-eq) under SSP5-8.5 in 2050 relative to 2018. These differences are even larger at the end of the century because urban expansion is projected to continue, though uncertainties also increase with longer projections. In 2100 relative to 2018, the estimated global RF is 0.00157 [0.00119,0.00255] W/m$^2$ (~1.68 [1.28, 2.73] Gt $CO_2$-eq) under SSP1-2.6, and it increases to 0.00187 [0.00144, 0.00288] W/m$^2$ (~2.00 [1.54,3.09] Gt $CO_2$-eq) under SSP2-4.5, and further increases to 0.00307 [0.00233, 0.00452] W/m$^2$ (~3.28 [2.50, 4.85] Gt $CO_2$-eq) under SSP5-8.5.

RF under all illustrative scenarios in both the near and far futures is not trivial for climate mitigation strategies. By using the equilibrium climate sensitivity tool, RF can be translated into the 100-year global warming of surface temperature. The RF in the near future 2050 relative to 2018 is equivalent to 0.00104 [0.00053,0.00177] °C of warming under the low emission scenario SSP1-2.6, 0.00107 [0.00057,0.00179] °C under the intermediate emission scenario SSP2-4.5, and 0.00128 [0.00068,0.00215] °C

under the very high emission scenario SSP5-8.5; similarly, the RF in the far future 2100 relative to 2018 equals 0.00127 [0.00063,0.00222] °C of warming under the low emission scenario SSP1-2.6, 0.00152 [0.00078,0.00259] °C under the intermediate emission scenario SSP2-4.5, and 0.00249 [0.00129,0.00420] °C under the very high emission scenario.

The estimated positive RF is primarily caused by the conversion of croplands, grasslands, savannas, and barren lands to urban lands in both the past and the future (Fig. 4; Supplementary Figs. 5, 6), which is consistent with that of albedo change. Croplands and cropland/natural vegetation mosaics together contribute the largest portion to future urban lands (35–40%) under all illustrative scenarios (Fig. 4; Supplementary Figs. 5, 6). That said, the amount of cropland-to-urban conversion may be underestimated, as other research has reported a higher percentage of new urban lands converted from croplands based on different land use/land cover products[36]. Moreover, it is known that croplands are underrepresented in Collection 6 of MODIS land cover product (MCD12Q1)[37], with many cropland parcels being mistakenly classified as savannas. For example, in southeast China, new urban lands are mainly projected to be converted from savannas, according to the MODIS 2018 base land cover (Fig. 2). Since savannas are absent in this climate region, these misclassified "savannas" are most likely short croplands mixed with trees. Barren lands also make a large contribution to total RF due to their much higher albedo compared to urban lands (Supplementary Fig. 4), despite their smaller share among all converted land types (Fig. 4). Among the biomes, desert and xeric shrubland biomes produce the largest albedo-induced RF for future urbanization effect (Supplementary Fig. 7). These biomes are characterized by a warm and dry climate; attention should be paid to the future urbanization and associated warming effects in these regions.

In sum, urbanization from 2001 to 2018 reduced the Earth's albedo and caused a warming effect; and future urbanization is expected to continue decreasing albedo which can induce a

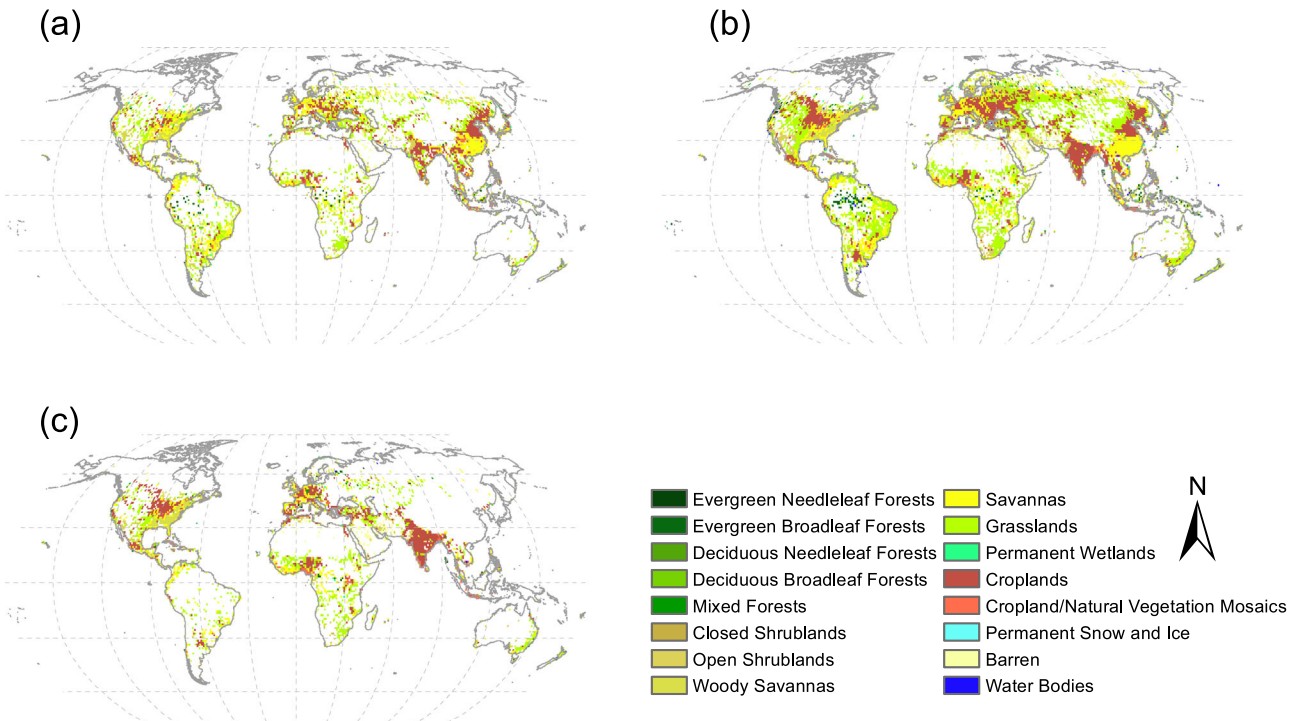

**Fig. 2 Dominant land cover converted to urban land.** The land cover types with the most area being converted into urban land at grid-level in each 1° grid **a** in 2018 relative to 2001, **b** in 2050 relative to 2018, and **c** in 2100 relative to 2050 under SSP2 scenario. See Supplementary Fig. 2 for results under SSP1 and SSP5 scenarios.

positive RF to the Earth's system and warm up the Earth's surface. The positive RF is mainly attributable to the replacement of higher albedo land covers, including croplands, grasslands, savannas, and barren lands with urban land with lower albedo. Note that our results are based on annual average albedo at a global scale, because there are substantial seasonal and spatial variations of albedo, studies at the city or regional scales using season-specific albedo (e.g., summer) can yield different results from this study (Supplementary Note 1). Our work provides empirical evidence of a warming effect caused by albedo changes in the course of urban expansion at a global scale, which could be used for comparison with future process-based modeling work. At the local city-scale, especially in these hotspot areas where albedo could be significantly reduced by urbanization, our study draws attention to the additional albedo effects from new urban land expansion on urban temperature, which could enhance the local heat island phenomena and negatively affect the increasing population that will live on there[1]. Our results may seem counter-intuitive to many people who think urban land is more reflective than adjacent natural land (mainly cropland), but this is not true for the current forms of urban land at a global scale, suggesting reflective and white materials are still not widely used in urban infrastructures and buildings. Our findings thus highlight the need to regulate and enhance the albedo of urban land.

**Future mitigation strategies**. The albedo of urban lands is more manageable than that of natural lands, providing us opportunities to mitigate warming effects. Our estimates of RF due to altered albedo are based on the observed albedo of contemporary urban lands. However, future urban landscapes may evolve with different morphology and construction materials as guided by planning policies and new construction techniques. One of the potential strategies is to replace conventional materials with reflective ones, such as cool roofs and reflective pavements. There is a potential to increase the albedo of current urban lands by 0.1

by using reflective and whiter materials for the pavements and roofs that currently typically constitute over 60% of urban surfaces[26]. Applying a conservative assumption that 1% of the global terrestrial area will be urban, and without considering the temporal and spatial variation of snow and radiation, Akbari et al.[26] estimated that an albedo increase of 0.1 in worldwide urban areas would result in a cooling effect that is equivalent to absorbing ~44 Gt of $CO_2$ emissions. Using the University of Victoria Earth System Climate Model and urban land distribution products of GRUMP and MODIS, they also found a long-term cooling effect that equals to a $CO_2$ reduction of 25–150 Gt of $CO_2$ through increasing urban albedo by 0.1 in temperate and tropical areas[27]. While these estimates vary substantially due to the use of different land cover classifications and uncertainties in model parameters, they do consistently suggest a strong potential to offset greenhouse gas emissions through manipulating urban land albedo. We caution that increasing urban albedo by 0.1 is challenging and may have unintended consequences on building energy consumption and thermal comfort[38], but various mitigation methods to manipulate albedo are worthwhile to explore. Another mitigation option is to construct more green roofs. Studies have shown that green roofs not only sequester substantial amounts of $CO_2$ through photosynthesis but also can significantly reduce the UHI effect in addition to providing many other ecosystem services[39,40]. Intriguingly, green roofs have higher albedo than many conventional black/brown roofs. Across the globe, roofs represent up to 32% of the horizontal surface of built-up areas[41], offering a great potential for offsetting warming effects with green roofs. However, one must simultaneously consider the impacts of green roofs on other biophysical processes systematically[42]. Increasing the percentage of non-roof urban green and blue (i.e., water) in new urban landscapes also can generally help reduce UHI and mitigate climate warming[43,44]. Other more general methods for albedo modification, such as

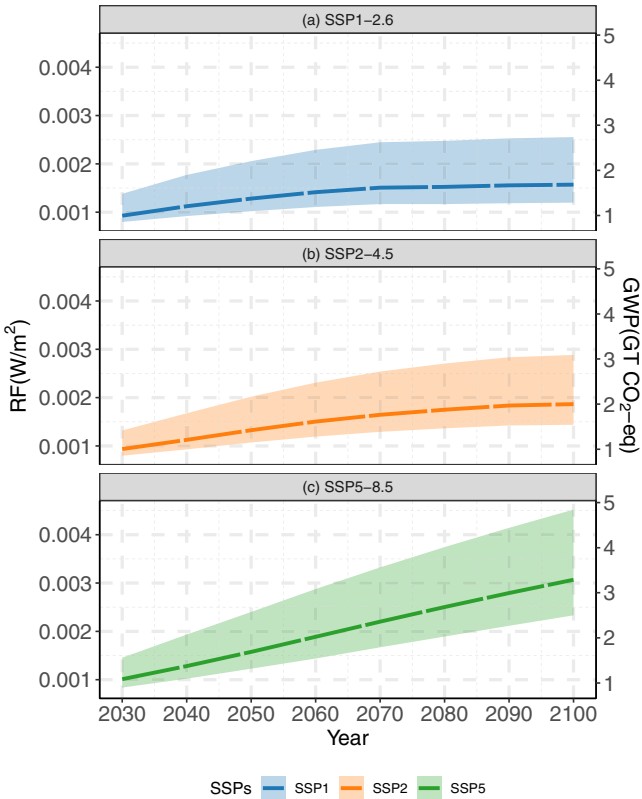

**Fig. 3 Top-of-atmosphere radiative forcing (RF, W/m²) due to projected urbanization.** The RF due to altered land surface albedo in 2030–2100 relative to 2018 for three illustrative scenarios: **a** SSP1-2.6, **b** SSP2-4.5, and **c** SSP5-8.5. The shaded area shows the 90% confidence intervals resulting from urbanization likelihood based on 100 urban expansion simulations.

artificial aerosol release into the urban atmosphere[45], may also be applied, although it would negatively affect the air quality.

**Uncertainties and limitations.** Climate impacts of urbanization are complex because other bio-geophysical and biochemical processes are coupled with the effects of albedo changes[8,18]. At regional and global scale, urbanization can reduce terrestrial net primary production[13,14,35], leading to increased release of soil carbon to the atmosphere[46], and resulting in less water infiltration and evapotranspiration[24,47]. These processes need to be examined together to determine the net climate impacts of urbanization. An empirically data-driven approach as adopted in this study can hardly address the net effect of all coupled processes. Future studies may consider using observation-constrained (e.g., carbon, water, and energy fluxes measured by urban eddy covariance sites) Earth system models to understand the net climate impact from urbanization due to altered bio-geophysical and biochemical processes. On the other hand, because these processes are coupled, other processes may affect the estimate of climate impacts of albedo changes. For example, the change in evapotranspiration during urbanization can affect the formation of clouds at local-to-regional scales, which in turn complicates the RF of albedo changes by altering blue-sky albedo and radiation transmission. In this study, the effect of cloudiness on transmission has been accounted for in the kernels but we have assumed stable proportions of diffuse and direct radiation for computing blue-sky albedo. We, however, argue that this effect has a limited impact on our estimate of RF, as if the pro-portion of direct and diffuse radiation were to significantly

change (thus influence the contribution of white-sky and black-sky albedo to the blue-sky albedo), the effect on the difference in the blue-sky albedo would be small, since the white-sky and black-sky albedo of each land cover type is similar (Supplementary Fig. 4). To support this argument, we conducted two experiments, where in the first one we artificially reversed the NCEP diffuse/direct radiation ratio of each grid for 2050 and 2100, and in the second one we used the projected ratios of diffuse and direct radiation from three CMIP5 models in 2050 and 2100. We found the resulting RFs were within 9% difference from our original estimation in the first experiment and within 4% difference in the second experiment in all illustrative scenarios (Supplementary Note 2). The first experiment represents extreme conditions that can rarely occur (i.e., monthly diffuse radiation is seldom larger than direct beam radiation). The second experiment represents more reasonable changes in diffuse/direct radiation ratios. Therefore, we argue that using constant diffuse/direct radiation ratios only causes minor uncertainties that are much less than uncertainties stemming from the use of different simulations of urbanization even within the same SSP scenario (Fig. 3).

Our estimates also inherit uncertainties from projections of urban land distribution that will likely be very different from the real path of urban development determined by future social and economic development, as well as regional and global urban planning policies. In this study, we mainly consider urbanization projections from three SSPs in our illustrative scenarios[4]. To increase our confidence in the estimates of a warming effect, we also estimated RF using a wider range of independent urbanization projections, including additional two SSPs (i.e., SSP3 and SSP4) from the same data source used in our illustrative scenarios above (hereafter Chen-2020)[4], four projections under the old SRES scenarios (hereafter Li-2017)[3], and one product without scenario description (hereafter Zhou-2019)[2](see methods for more details). In general, the estimated RF using Li-2017 is much higher than those of Zhou-2019 and Chen-2020 (Fig. 5). The variations in the estimation signify the uncertainties in the warming effects as induced by different urbanization paths. Nevertheless, the estimated RF is positive across all possible combinations of urbanization projection scenarios with RCP emission scenarios, suggesting a robust and consistent warming effect regardless of varying urbanization projections.

RF has been widely used and proved to be an effective tool for assessing the influence on global mean temperature of most individual agents that can affect the Earth's radiation balance[48,49], although it is not a perfect tool. One limitation of RF is its nonadditivity of different RF agents, but this is not a problem for this study as we exclusively focused on RF due to albedo changes. Improved understanding has suggested that the recently proposed concept of effective RF (ERF) may be a better indicator of the temperature response for some forcing agents[50], because ERF considers other rapid adjustments of the troposphere (e.g., atmospheric temperatures, water vapor, and clouds) in addition to stratospheric temperature, which is the only adjustment for RF. However, the method for calculating ERF remains unsettled, ERFs require more computational resources than RF and spread wider across models due to rapid model-dependent feedbacks, and ERFs can have difficulties in isolating small forcing due to uncertainties relative to the forcing itself. Still, future work should consider adopting ERFs for better estimating albedo-change-induced RF when ERF radiative kernels become available.

## Methods
**Land cover**. Urban landscapes are characterized by small clusters of patches, forming land mosaics that are distinct from natural landscapes. An accurate esti-mation of climate forcing requires a land cover dataset at high resolutions that does

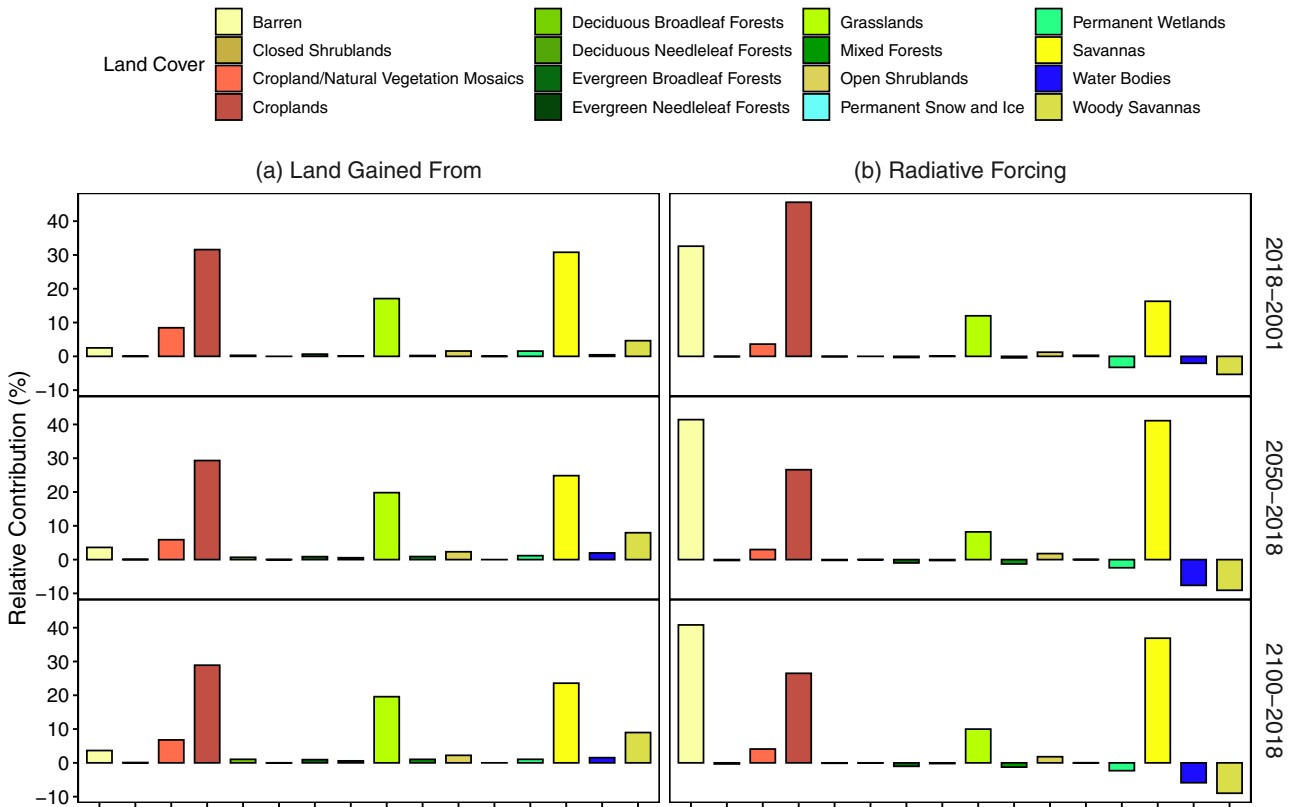

**Fig. 4 Contributions of different land cover types to gained urban land and radiative forcing. a** Contribution of different land covers to the global total new urban land areas; and **b** contribution of different land covers to the global total albedo-change-induced warming potential in 2018 relative to 2001(2018–2001), in 2050 relative to 2018 (2050–2018), and in 2100 relative to 2018 (2100–2018). Data for future periods are based on the SSP2-4.5 scenario. Results of SSP1-2.6 and SSP5-8.5 scenarios are similar to that of SSP2-4.5 and are provided in Supplementary Figs. 5 and 6.

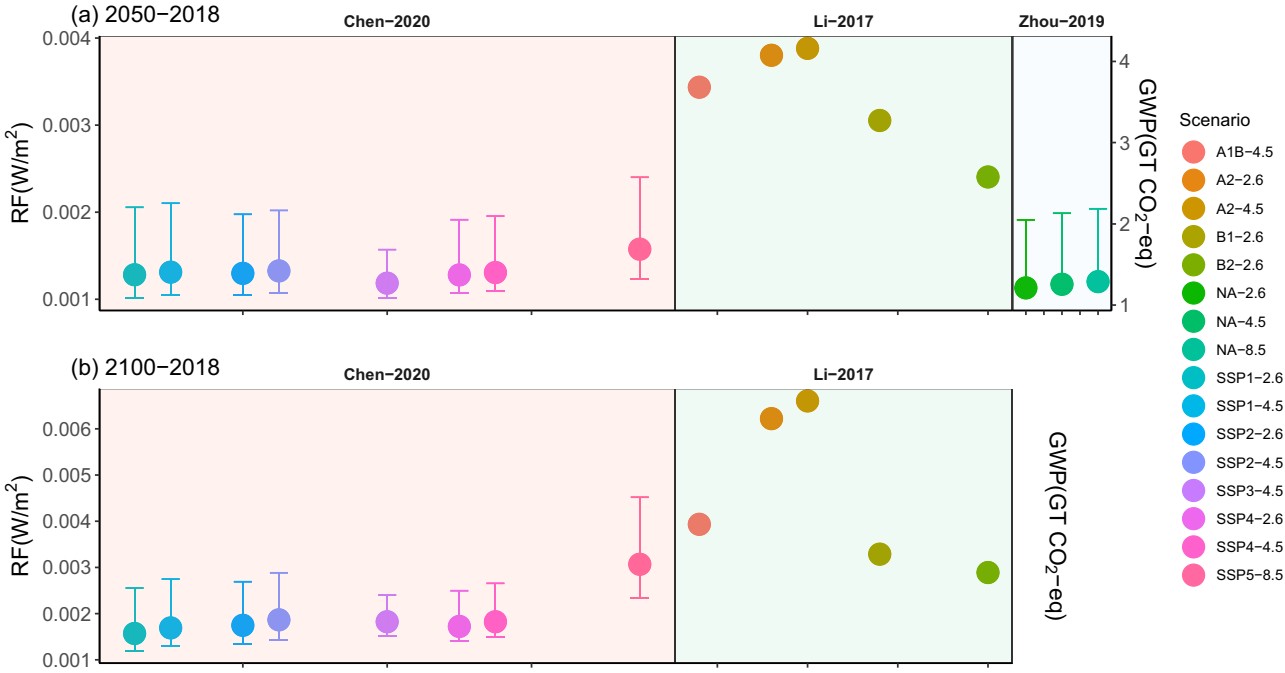

**Fig. 5 Comparisons of radiative forcing among using different products of urban land projections.** Comparisons of albedo-induced radiative forcing (RF, W/m²) and global warming potentials (GWP, Gt CO2-eq) due to urbanization using different products of high-resolution (1-km) urban land projections **a** in 2050 relative to 2018 and **b** in 2100 relative to 2018. The compatibility between SSPs (Chen-2020) and RCPs was adopted from Riahi et al.[53], and the compatibility between SRES (Li-2017) and RCPs was determined based on the limits of global warming (for example, B1 matches RCP2.6 since the global warming limits of B1 is 1.1–2.9 °C). "NA" was used for the urbanization scenario in Zhou-2019 as this product follows no scenario families. Error bars shows 90% confidence intervals.

not omit small urban patches. In this study, the RF estimates are based on 500-m and 1-km land cover datasets. This fine resolution is necessary to preserve spatial details of small urban patches while avoiding the large underestimation of urban land areas at coarse resolution (e.g., ~19% underestimation at 10 km compared to that at 1 km)[3]. We used 500-m resolution MODIS Land Cover product (MCD12Q1v006) for historical land cover changes. For future urban land cover distributions, we used the global urban land expansion products simulated under the SSPs for 2030–2100 (i.e., Chen-2020)[4]. The simulation performance was tested using historical urban expansion from 2000 to 2015 based on Global Human Settlement Layer[51], where the agreement between simulated and observed urban land was evaluated using the Figure of Merit (FoM) indicator[52] that has showed similar or better values than those reported in other existing land simulation applications[4]. The high-resolution Chen-2020 also shows very high spatial consistency with the prominent coarse resolution global urban land projection LUH2 that is recommended in CMIP6[4]. Considering different scenarios is also necessary to account for the uncertainties of future socioeconomic and environmental conditions, so we included simulated urban lands under three scenarios (Supplementary Table 1): Sustainability -SSP1, Middle of the Road - SSP2, and Fossil-fueled Development - SSP5[53]. Within each SSP scenario, the product provides a likelihood map of each grid becoming urban, based on 100 urbanization simulations. We used the likelihood map to account for spatial uncertainties of urban expansion by deriving 90% confidence intervals of projected urban land demand within a SSP scenario. We used the MODIS IGBP Land Cover classes (Supplementary Table 2) and resampled the original 500-m resolution MODIS products in 2018 to 1-km resolution to match the future simulations when it was used as a baseline year. To isolate the independent effect of urbanization (vs other types of land uses) in future estimates, land covers that are not converted to urban are assumed to have the same cover types as in 2018 (i.e., the baseline year). Though there are other global land cover products for current periods, we choose the MODIS IGBP land cover products because the albedo look-up maps (LUMs) were based on IGBP land cover types (see Albedo Look-Up Maps).

To further evaluate the uncertainties caused by different projections of future urbanization, we also included the other two SSPs from Chen-2020, and another two 1-km resolution urban land cover products projected for the future for the purpose of comparison. The other two products include four projections of SRES scenarios (i.e., A1, B1, A1B, and B2) (i.e., Li-2017 mentioned above)[3] and one without scenario description but assumed historical development would continue (i.e., Zhou-2019 mentioned above)[2]. These projections of future urban land expansion were calibrated with different historical urban land products and can be regarded as independent.

**Albedo look-up maps (LUMs).** Albedo Look-Up Maps (LUMs)[31] were derived from the intersection of MODIS land cover[54] and surface albedo[55] products, which are used to determine the albedo values for each IGBP land cover type by month and by location. Monthly means of white-sky (i.e., diffuse surface illumination condition) and black sky (i.e., direct surface illumination condition) during 2001–2011 were processed for snow-free and snow-covered periods for each of the 17 IGBP land cover classes at spatial resolutions of 0.05°–1°[31]. The LUMs have been verified by comparing the reconstructed albedo using the LUMs with the original MODIS albedo, which shows very similar values[31]. We used the LUMs at a resolution of 1° due to the significantly fewer missing values, to assure the spatial continuity of albedo changes at a global scale while keeping the matches with the 1° resolution of radiation data and RF kernels. The underlying assumption is that albedo of the same land cover type varies insignificantly within a 1° grid.

**Snow and radiation product.** Snow cover can significantly change the albedo of land regardless of cover types (Supplementary Fig. 4). In this study, we tally monthly albedo using snow-free and snow-covered categories in estimating RF. Past and present snow-free and snow-covered conditions were derived from level 3 MODIS/Terra Snow Cover (MOD10CM.006)[56] at 0.05° spatial resolution and resampled to a 1° spatial resolution. Monthly means of 2001–2005 vs 2015–2019 were used for 2001 and 2018 respectively. For future periods, ensemble mean snow cover for each year and month, projected under the CMIP5 framework for three Representative Concentration Pathway (RCP) scenarios (i.e., RCP2.6, RCP4.5, and RCP8.5) were used (for more details see Supplementary Note 2B). By comparing the model outputs with MODIS observations for a recent decade (2006–2015), we found that the multi-model mean snow cover was systematically biased compared to MODIS observations. Consequently, we calibrated the ensemble mean projections by subtracting the biases for the grids. In each 10th year of the future (e.g., 2030, 2040, etc.), the decadal monthly mean snow cover (e.g., 2026–2035 for 2030, and 2036–2045 for 2040, etc.) was used for the year.

We used the long-term monthly averages (1981–2010) of diffuse and direct incoming surface solar radiation reanalysis Gaussian grid product from National Centers for Environmental Prediction (NCEP)[57]. Visible and near infrared beam downward radiation and diffuse downward radiation from NCEP were used to compute the white-sky and black-sky fractions. As for snow cover, ensemble mean shortwave radiation at surface ($SW^{SF}$) and at top-of-atmosphere ($SW^{TOA}$) projected from CMIP5 models (Supplementary Note 3C) for RCP2.6, RCP4.5, and RCP8.5 were collected for empirically computing future albedo kernels (see section 3.4 below).

**Radiative kernels.** Radiative kernels were used to compute top-of-atmosphere RF due to small perturbations of temperature, water vapor, and albedo. We used the latest state-of-the-art albedo kernels calculated with CESM v1.1.2[58] to compute RF in 2018 relative to 2001. In brief, the albedo kernel is the change in top-of-atmosphere radiative flux for a 0.01 change in surface albedo. The CESM1.1.2 kernels are separated into clear- and all-sky illumination conditions. We used the all-sky kernels because we include both black-sky and white-sky albedos. For future periods, because there are no available radiative kernels produced from general circulation models, we approximated the future kernels using an empirical parameterization following Bright et al.[59]:

$$K_m(i) = SW^{SF}(i) \times sqrt\left(\frac{SW^{SF}(i)}{SW^{TOA}(i)}\right) / (-100) \quad (1)$$

where $m$ is the month, $i$ is the location, and $SW^{SF}$ and $SW^{TOA}$ are the surface and top-of-atmosphere shortwave radiation; dividing by $-100$ is for matching the CESM1.1.2 kernel definition of a 0.01 change in surface albedo.

**Estimation of albedo change and RF.** We analyzed the RF in 2018 due to albedo changes caused by urbanization since 2001 (2018–2001), and in the future from 2030 to 2100 at decadal intervals (i.e., 2030, 2040, 2050, ..., and 2100) since 2018 under three illustrative scenarios: SSP1-2.6, SSP2-4.5, and SSP5-8.5, which combine SSP-based urbanization projections and RCP-based climate projections. The three illustrative scenarios were selected following the scenario designation of the latest IPCC report[50] and represent low greenhouse gas (GHG) emissions with $CO_2$ emissions declining to net zero around or after 2050, intermediate GHG emissions with $CO_2$ emissions remaining around current levels until the mid-century, and very high $CO_2$ emissions that roughly double from current levels by 2050, respectively. We selected 2018 as the baseline year to divide the past from the future because 2018 was the latest year with available MODIS land cover products at the time of this study. We used ArcGIS 10.6 to produce spatial maps of all variables, including area of each land cover type within a 1° × 1°-grid, snow cover, albedo, radiation, and kernels, and R 3.6.1 to compute the RF.

We focused only on albedo changes induced by urbanization, including the conversions from all other 16 IGBP land cover types to urban land. The changes of albedo for each grid $(x, y)$ of a month $(m)$ were obtained by computing the difference between albedo of that grid in the baseline year $(t = t_0)$ and in a later year $(t = t_1)$ with urban expansion:

$$\triangle\alpha_{m,t1-t0}(x,y) = \alpha_{m,t=t1}(x,y) - \alpha_{m,t=t0}(x,y) \quad (2)$$

where $\alpha_{m,\,t=t1}(x,y)$ and $\alpha_{m,\,t=t0}(x,y)$ is the albedo for each grid $(x,y)$ of a month $(m)$ at the base year and later year respectively; the grid-scale albedo is computed as the weighted sum of albedo by land cover types with the weighing factor corresponding to areal percentage of a land cover within the grid. The albedo for each land cover type of a grid was then obtained by applying the albedo LUMs that provide spatially continuous black-sky, white-sky, snow-covered, and snow-free albedo maps for a given month for each land cover. Firstly, monthly mean albedo is computed as:

$$\alpha_{m,t}(x,y) = \sum_{l=1}^{17}\sum_{s=0}^{1}\sum_{r=0}^{1} f_{l,t}(x,y)f_{s,m,t}(x,y)f_{r,m,t}(x,y)\left(\alpha_{l,s,r,m}(x,y)\right) \quad (3)$$

where $m$ is the month, $t$ is the year, $l$ is the land cover type, $f_l$ is the proportion of a cover type within the grid, $f_{s,m,t}$ is the fraction for snow-covered ($s = 0$) and snow-free ($s = 1$) conditions of the time $(m, t)$, $f_{r,m,t}(x, y)$ is the fraction for white-sky ($r = 0$) or black-sky ($r = 1$) conditions of the time, and $\alpha_{l,s,r,m}(x, y)$ is the albedo for land cover type $l$ in month $m$ that is extracted from the albedo LUMs corresponding to snow condition ($s$) and radiation condition ($r$). The annual mean albedo change is reported as the mean of monthly albedo change:

$$\triangle\alpha_{t1-t0}(x,y) = \frac{1}{12}\sum_{m=1}^{m=12}(\alpha_{m,t=t1}(x,y) - \alpha_{m,t=t0}(x,y)) \quad (4)$$

The conversion of other land covers to urban land can contribute differently to the global RF, as the total area that is converted into urban land is different among non-urban land covers and the albedo differences between urban land and non-urban land cover types vary. To estimate the proportional contributions of different land conversions, we first decomposed the total albedo of each grid into the proportion of each land cover type:

$$\alpha_{l,m,t}(x,y) = f_{l,m,t}(x,y)\sum_{s=0}^{1}\sum_{r=0}^{1} f_{s,m,t}(x,y)f_{r,m,t}(x,y)\left(\alpha_{l,s,r,m}(x,y)\right) \quad (5)$$

The global RF due to albedo change caused by conversion from each non-urban land cover type ($l \neq 13$) to urban land ($l = 13$) (see Supplementary Table 2 land cover labels) was calculated as:

$$RF_{\triangle\alpha,l(l\neq13),global} = \frac{1}{A_{Earth}}\sum_{i=1}^{n}\sum_{m=1}^{12}(\alpha_{13,m,t=t1}(i) - \alpha_{l,m,t=t0}(i))\Delta p_{l\to13}(i)Area(i)K_m(i) \quad (6)$$

where $i$ refers to a grid, $n$ is the total number of pixels on global lands, $A_{Earth}$ is the global surface area (5.1 × 10⁸ km²), $\alpha_{13,m,t=t1}(i)$ is the albedo of urban land in month $m$ in the later year with urban expansion, $\alpha_{l,m,t=t0}(i)$ is the albedo of a targeted non-urban land cover type in the base year $t_0$, $\Delta p_{l\to13}$ is the percentage of

the non-urban land cover type that is converted to urban land in the year $t_1$ compared to year $t_0$, $Area(i)$ is the area of the pixel, and $K_m(i)$ is the radiative kernel at the grid.

The global RF due to urbanization-induced albedo changes was then calculated as:

$$RF_{\triangle\alpha,global} = \sum_{l=1}^{17} RF_{\triangle\alpha,l,global}(l \neq 13) \qquad (7)$$

**GWP: $CO_2$-equivalent**. We followed GWP calculations by explicitly accounting for the lifetime and dynamic behavior of $CO_2$ to convert RF to $CO_2$ equivalent[60,61]:

$$GWP[kg\ of\ CO_2 - eq] = \frac{\int_{t=0}^{TH} RF_{\triangle\alpha,global}(t)}{k_{CO_2}\int_{t=0}^{TH} y_{CO_2}(t)} \qquad (8)$$

where $k_{CO2}$ is radiative efficiency of $CO_2$ in the atmosphere (W/m²/kg) at a constant background concentration of 389 ppmv, which is taken as $1.76 \times 10^{15}$ W/m²/kg[62], and $RF_{\triangle\alpha,global}$ is the global RF caused by albedo changes (W/m²). $y_{CO_2}$ is the impulse-response function (IRF) for $CO_2$ that ranges from 1 at the time of the emission pulse ($t=0$) to 0.41 after 100 years, and here it is set to a mean value of 0.52 over 100 years[60]. The time horizon (TH) of our GWP calculations was fixed at 100 years following IPCC standards and previous studies[60,63,64].

**Global mean surface air temperature change**. We estimated the 100-year global mean surface temperature change for the estimated RF by adopting an equilibrium climate sensitivity (ECS), defined as the global mean surface air temperature increase that follows a doubling of pre-industrial atmospheric carbon dioxide (RF = 3.7 W/m²). Given a value of RF induced by a forcing agent, the temperature change is estimated as RF/3.7 × ECS. To consider the uncertainties of ECS, we adopted a mean value of 3 °C and a very likely (90% confidence interval) range of 2–5 °C following IPCC AR6[50]. Without knowing the exact distribution shape of ECS and future albedo-change-induced RF, we created a log-normal distribution (Supplementary Note 4) to approximate their asymmetric distribution through numerical simulation. We then conducted Monte Carlo simulations that draw 5000 random samples from each distribution to jointly estimate the uncertainties of global mean surface air temperature changes. We report the mean and 90% interval ranges of the change in temperature.

## Data availability

This study makes use of publicly available data. The MODIS land use land cover products in 2001 and 2018 are available from https://lpdaac.usgs.gov/products/mcd12q1v006/. MODIS snow cover data are available from https://nsidc.org/data/MOD10CM/versions/6. The future projection of urbanization under different SSPs is available from https://doi.pangaea.de/10.1594/PANGAEA.905890. The future projection of urbanization using old SRES scenarios is available from http://geosimulation.cn/GlobalLUCCProduct.html. Zhou-2019 future urbanization projection is available from https://doi.org/10.26188/5c2c5d0ed52d7. NCEP radiation reanalysis data are available from https://psl.noaa.gov/data/gridded/data.ncep.reanalysis.derived.otherflux.html. CMIP5 climate projections including snow cover, surface shortwave radiation, top-of atmosphere shortwave radiation, and diffuse radiation are available from https://esgf-node.llnl.gov/projects/esgf-llnl. The CAM5 radiative kernels for albedo are available from https://doi.org/10.5065/D6F47MT6. The albedo LUM data are available from https://doi.org/10.15482/USDA.ADC/1523392[65]. Source data for figures are provided in ref. [66].

## Code availability

Example codes written in R for computing RF were provided on GitHub https://github.com/ouyangzt/Urban_Albedo_RF and ref. [66], and more detailed codes will be made available upon request by contacting the corresponding authors.

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

## Acknowledgements
We acknowledge Feng Gao from USDA for providing the data of MODIS albedo look-up maps. We thank Raul R. Cordero for providing constructive suggestions that improved the study. S. F. gratefully acknowledges the support of ANID (Preis 1191932 & REDES180158) and CORFO (Preis 19BP-117358, 18BPE-93920, 18BPCR-89100 & 17BPE-73748). This study was funded in part by the NASA LCLUC program (80NSSC20K0410). P.F. also acknowledges the support from the NASA LCLUC program for SENA (NNX15AD51G) and (80NSSC20K0740).

## Author contributions
Z.O. and J.C. conceptually designed the study. Z.O. performed the main data analysis and drafted the manuscript. P.S., T.J, S.F., C.L., F.L., R.J., P.F., X.L, C.A.W., G.C, and C.W. provided and analyzed data, and revised the manuscript.

## Competing interests
The authors declare no competing interests.
