## [Peer Review File · Nature Communications]

Albedo Changes Caused by Future Urbanization Contribute to Global WarmingREVIEWER COMMENTS

Reviewer #1 (Remarks to the Author):

Planetary albedo changes caused by future urbanization will warm the earth

Using the instantaneous shortwave radiative forcing at the top-of-the-atmosphere (TOA) as a global climate forcing metric ("RF_TOA"), the study investigates the climate forcing of recent historical, mid-21st century, and end 21st century surface albedo changes associated with urban expansion (or "urbanization"). The authors rely on MODIS land cover and surface albedo products (or their derivatives) along with datasets of projected urbanization and snow cover linked to several SSP-RCP scenarios from a variety of sources. For the most part the work appears thorough and technically sound, and the authors have made sensible use of key datasets to address an important knowledge gap surrounding the climate forcing of urban expansion. Given the chosen climate forcing metric, their conclusions appear sufficiently supported by their analysis. However, the focus solely on surface albedo change is not sufficiently justified, and the chosen metric provides an incomplete and potentially misleading picture of the climate forcing attributable to urban expansion. Land cover changes driven by urban expansion affect surface fluxes of heat and moisture in parallel, resulting in perturbations to atmospheric properties that can lead to additional radiative flux perturbations locally (which can either dampen or enhance the albedo-change induced instantaneous shortwave RF at TOA) as well as trigger remote responses. The comparison of the local albedo change-driven forcing to the global mean forcing from CO2 emissions is misleading at best and counter-productive at worst.

The manuscript (especially the title and Introduction paragraphs) seem to be unjustly overstating the significance of the results to make the manuscript Nature-worthy, and many statements need to be revised and appropriately caveated. These caveats and limitations notwithstanding, however, I do think there is value to the empirically-based albedo change estimates that can serve as an important reference/benchmark for the climate and UHI modeling community. I will leave it to the Editor to decide if a toned-down interpretation of the results would be of interest to the general audience of Nature Communications.

Major comments:

Scoping

The authors make no sincere effort to describe the physical processes and mechanisms perturbed at the surface by land cover conversions to urban lands, and how these in turn connect to perturbed atmospheric process driving climate changes across different spatial and temporal scales. Few links to the scientific literature on this front are presented, and thus the singling out of surface albedo change appears not sufficiently well-grounded and justified as a result. Several studies (Zhao et al. 2014; Georgescu et al. 2009a) point to the relevance and importance of surface albedo changes (reference info is provided at the end of this review) in the context of urbanization and local direct temperature effects -- which the authors could draw upon for improved contextualization and justification. On a similar note, no real effort is made to establish the credibility of the chosen RF-based measure as a suitable indicator of the ensuing climate response (i.e., temperature change) likely to be experienced both locally and remotely. The motivation for moving away from the instantaneous towards "effective" radiative forcing (see IPCC 5AR WG1 Chapter 8; Sherwood et al. 2015; Forster et al. 2016) is that it is a better indicator of the temperature response since the atmospheric adjustments are included (see Georgescu et al. 2009b for an example of rapid atmospheric adjustments in response to surface energy balance changes linked to urban expansion). Further, and on a related note, the authors also make no real effort to discuss how urban expansion might affect climate remote of the urbanized areas, which is important from the regional-to-global adaptation/mitigation perspective. This is also exemplified in Georgescu et al. 2009b. At the global scale, Jacobson & ten Hoeve (your

ref. 18) clearly demonstrate that remote responses are larger than the local response and dominate the global mean response to urbanization.

A review of the UHI-climate modeling or land use/land cover change climate modeling literature through this lens might help the authors qualitatively define some confidence bounds for their chosen RF measure in terms of its robustness as an indicator climate change, both locally and globally. Caveats and limitations are likely to emerge naturally from a more honest and comprehensive review of the literature in these areas.

Methods

The appropriateness of the "CO₂-eq." measure notwithstanding, its calculation is poorly described, and the same results are presented in some places with units "Gt-CO₂-eq.," while in others with units "Gt-CO₂-eq./yr". By my estimate the "per yr" values are two orders of magnitude too high for the reported RF_TOA results. Further, the authors acknowledge a changing future radiation regime as being "important" and "significant" (P7, L250-251), but do not factor this into their analysis. Embedded in the radiative kernels applied to quantify radiative forcing ("RF_TOA") in all future time steps (up to 2100) is the atmospheric state of 2006/2007 (i.e., clouds, aerosols, water vapor, and other factors affecting shortwave radiative transfer), but no justification is given as to their suitability in estimating radiative forcings under future atmospheric conditions. Jonko et al. (2012) for example show substantial deviations (up to ca 20%) in albedo kernels between those estimated under 8 x CO₂ forcing and those with the present atmospheric background state. The use of the 2006/2007 Pendergrass kernel (ref 36) to estimate forcings in 2050 and 2100 is thus highly questionable. The better approach would be to apply a kernel (or radiative transfer model) that can account for changes to an evolving atmospheric background state. An example is the model underlying the kernel of Bright & O'Halloran (2019) that appears as a simple function of surface downwelling solar radiation flux – a variable which is freely available in a variety of CMIP5 & 6 archives (alternatively the simplified kernel model of Donohoe et al. (2020)). The authors appear to have relied on such archives for future snow cover, so an extension to included future radiation budget variables as input is not an unreasonable amount of added effort.

Other general comments

The English grammar needs improvement.

Other specific comments

P1, Title: I suggest an alternate title without the word "warm" that is more faithful to the content and scoping and which is more indicative of the actual metric applied in the work (e.g., instantaneous SW RF).

P2, L45, Abstract: Please refrain from referring to positive and negative RF results as "warming" and "cooling" here and elsewhere throughout the manuscript. "Warming" and "cooling" imply that some temperature indicator has been investigated, which has not.

P3, L55: "Land cover and land use change (LCLUC)". Please consider aligning with the prevailing acronym of the scientific literature: Land use/land cover change (LULCC).

P3, L 57-58: Possible to indicate what the current global urban area occupation is? It would help the reader acquire an immediate feel for the scale of the research problem being addressed.

P3, L58-60: I think it is enough to just say "physical properties" here instead of the long and seemingly arbitrary list of examples.

P3, L63-64: The preceding sentence mentions several climate forcing pathways or

mechanisms, so it is unclear why there is a sudden shift here towards the focus on albedo. A justification for this shift in focus is needed, since the likelihood of a reader not having a background in the subject matter is higher for a more generalist journal like Nat. Comm. In fact, the merit of this entire study hinges on the relevance of surface albedo, and thus it needs to be sufficiently demonstrated that surface albedo changes in fact have high relevance. See the first major comment above surrounding the article scoping/framing.

P3, L65-68: This sentence needs re-writing to improve clarity and technicality. How does "albedo, evapotranspiration, and heat conduction" "affect local-to-global climate" as measured with the metric "radiative forcing (RF)"? Surface "albedo" is a state property, but RF is a measure of some state change (i.e., albedo change). Similarly, "evaporation" and "heat conduction" are fluxes characterizing the state of the system, and changes to these do not directly perturb the planetary energy balance (i.e., do not contribute to RF).

P3, L68: Please provide more details about this important reference in the reference list which appears only as "IPCC Climate Change 2014". I assume the authors are referring here to the 5th Assessment Report – specifically the work of Working Group I – and specifically the radiative forcing chapter (Chapter 8, Myhre et al. 2013)?

P3, L73: Replace "cooling effect" with "negative radiative forcing"...also L76 and elsewhere.

P4, Section 2: Consider re-naming this section since the main results appear to be presented in the previous (and last) sentence of Section 1. Alternately consider presenting the results of L122-124 somewhere in Section 2 instead.

P4, L148: I suggest to use alternative terminology and acronym to "surface radiative forcing" for surface level radiative flux perturbations, as "radiative forcing" is conventionally reserved for an external (i.e., planetary) energy balance perturbation. How about "surface shortwave forcing" with acronym "DeltaSW_SUF" or something.

P4, L128: What is meant by surface radiative forcing "changes"? RF denotes a change (or difference) in radiative flux with respect to some reference state. Is the reference albedo state here the flux associated with 2001 albedo, i.e., $RF = f(\text{alb}_{2018} - \text{alb}_{2001})$, in which case the word "change" is not needed? Or has an RF been defined also for 2001 (e.g. albedo change from pre-industrial?), and if so, what is the reference year that is used?

P14, Figure 2: It is impossible to see differences between the right and left column of results (i.e., 2050-2018 vs. 2100-2028 results). It would be more informative if the right column instead showed the 2100 differences with respect to the 2050 results. As for Figure 1, consider also going with a different map projection that maximizes the area(s) of interest, i.e., the land areas between -55°S and 70°N. Or consider cropping out unnecessary latitude and longitude bands.

P5, L138: You mean local forcing?

P5, L164: "albedo-induced warming effect". Not to beat a dead horse, but again this terminology is incorrect and misleading, as the metric under scope is the (instantaneous shortwave) RF at TOA (in W/m^2), and it is unknown what the temperature response at the surface is, which might be opposite to the instantaneous SW RF at TOA, particularly areas remote of where the urbanization occurs (see your refs. 4 & 18 and Section 2.4).

P6, L206-210: The CO₂-eq. results here (and in the Abstract and on P3, L122-124) are given with units "per year", but they appear similar in magnitude to those presented on P5, L167-173 and in Figure 5 for the same magnitude RF which are NOT given on a "per year" basis. Please explain the discrepancy (and see my comment below on Section

4.4).

P7, Section 2.4: The authors mention that the state of the present day atmosphere – specifically the monthly shares of diffuse and direct solar radiation - is extrapolated to the future when applied to estimate future albedo changes associated with future urbanization. They acknowledge that this “may change significantly” in the future, but yet that this would not affect their future albedo calculations since the direct and diffuse albedos are “similar”. The authors however do not discuss the implications that this same assumption has on their future RF_TOA estimates, calculated here using a radiative kernel based on a 2006/2007 atmospheric state. Albedo kernels are sensitive to clouds, aerosols, and water vapor (affecting shares of direct vs. diffuse radiation), so the authors should either demonstrate that the consequence of this decision has little impact on the results, or consider modifying their approach.

P8, L280: Are the authors sure that these SSPs are compatible with all three RCPs investigated? See for example Kebede et al. (2018). It is not made clear in the methods which SSPs are combined with the three RCPs, although an inspection of Figure 5 suggests that compatibility considerations have been ignored.

P9, L 313: Why this older v005 product and not the latest MODIS snow cover product (v006) whose accuracy has been demonstrably and notably improved, notably due to improvements in the snow detection algorithm? (see Masson et al. 2018). Also, what variable from the MOD10CM product is being used? I assumed fractional snow cover (FSC) upon first reading but am now in doubt when looking at Eqs. 2 & 3 below which seem not to make use of FSC but some binary flag instead (either 1 for snow-free or 0 for snow), in which case this should be stated clearly. If FSC is in fact used in Eqs. 2 & 3, then these equations appear incorrectly formulated, as FSC should be used to weight the monthly snow-free and snow-covered albedos sourced from the look-up maps.

P9, L320: “biased in MODIS observations”? Please correct.

P11, Section 4.4: The units provided here for Eq. (9) are “Gt-CO2-eq.” as well as those for the results presented in Figure 5. But the same results appear to be presented in the Abstract as “Gt-CO2-eq./yr” which is a two-orders-of-magnitude discrepancy when summed over TH = 100 years. What is being integrated to TH in the numerator of Eq. (9)? The relationship between variable “RF_delta_alb” and time (“t”) is unclear. Are the transient (or time-dependent) RF_d_alb pathways presented in Figure 3 being used here? Or does “RF_d_alb” correspond to the albedo differences between 2050 (or 2100) and 2018 – and if so – is this value being multiplied by TH? Please improve the description of the CO2-eq. calculation methodology, and harmonize variable naming to be consistent across equations. An annual time step is absent from all “RF” and “d_alb” equations of sections 4.1 and 4.2.

P13, Figure 1. Differences between panels (c) and (b) are imperceptible. Please consider showing instead the difference from 2100-2050 as panel (c) so that the additional change becomes apparent.

P14, Figure 2. Similar comment as above – differences between panels (b) and (a) are imperceptible. The same goes for the differences between panels (d) and (c). Please consider an alternate way of presenting the differences between 2100 and 2050.

References

Jonko et al 2012. Climate Feedbacks in CCSM3 under Changing CO2 Forcing. Part I: Adapting the Linear Radiative Kernel Technique to Feedback Calculations for a Broad Range of Forcings, *Journal of Climate*, 25, 5260-5272, 10.1175/jcli-d-11-00524.1.

Zhao et al. 2014. Strong contributions of local background climate to urban heat islands, *Nature*, 511, 216-219, 10.1038/nature13462.

Georgescu et al 2009a. Climatic effects of 30 years of landscape change over the Greater Phoenix, Arizona, region: 1. Surface energy budget changes, Journal of Geophysical Research: Atmospheres, 114, <https://doi.org/10.1029/2008JD010745>.

Georgescu et al. 2009b. Climatic effects of 30 years of landscape change over the Greater Phoenix, Arizona, region: 2. Dynamical and thermodynamical response, Journal of Geophysical Research: Atmospheres, 114, <https://doi.org/10.1029/2008JD010762>.

Sherwood et al. 2015. Adjustments in the Forcing-Feedback Framework for Understanding Climate Change, Bulletin of the American Meteorological Society, 96, 217-228, [10.1175/bams-d-13-00167.1](https://doi.org/10.1175/bams-d-13-00167.1).

Forster et al. 2016. Recommendations for diagnosing effective radiative forcing from climate models for CMIP6, Journal of Geophysical Research: Atmospheres, 121, 412,460-412,475, [10.1002/2016jd025320](https://doi.org/10.1002/2016jd025320).

Bright & O'Halloran 2019. Developing a monthly radiative kernel for surface albedo change from satellite climatologies of Earth's shortwave radiation budget: CACK v1.0, Geosci. Model Dev., 12, 3975-3990, [10.5194/gmd-12-3975-2019](https://doi.org/10.5194/gmd-12-3975-2019).

Donohoe et al. 2020. The Effect of Atmospheric Transmissivity on Model and Observational Estimates of the Sea Ice Albedo Feedback, Journal of Climate, 33, 5743-5765, [10.1175/jcli-d-19-0674.1](https://doi.org/10.1175/jcli-d-19-0674.1).

Masson et al 2018. An Assessment of Existing Methodologies to Retrieve Snow Cover Fraction from MODIS Data, Remote Sensing, 10, 619.

Kebede et al 2018. Applying the global RCP-SSP-SPA scenario framework at sub-national scale: A multi-scale and participatory scenario approach, Science of The Total Environment, 635, 659-672, <https://doi.org/10.1016/j.scitotenv.2018.03.368>.

Reviewer #2 (Remarks to the Author):

Major comments

Radiative kernels are here used to compute TOA RF due to albedo changes. The authors use CESM1.1.2 tool to make the calculation. The authors do not explain if this tool also take into account the associated changes in evapotranspiration. I understand that it is not the case. In my opinion, the study has to show the impact on evapotranspiration and potentially evaluate the uncertainties due to cloudiness changes. Moreover, the authors should also add a life cycle analysis (i.e., to build new urban areas potentially generates important greenhouse gas emissions).

The study does not discuss the impact on carbon fluxes - to remove vegetation areas potentially leads to important changes of carbon sequestration by the surface.

It is not clear how snow covers the urban area. Is there a fraction of snow cover with a snow depth?

Finally, it seems that the authors have used a stable solar radiation. They argue that the effect of solar radiation variation on their estimates are limited, as the change of total solar radiation is slow. Please could the authors show solar radiation changes in the different climate projections? Could they show the change also between the direct and diffuse components? And could they discuss the impact of an albedo increase on the increase of multi-scattering contribution bounced back to the surface.

To conclude, the study is very interesting, and conclusions are original. However, the potential impacts on other bio-physical processes are not enough analyzed and discussed (in my opinion).

Minor comments

- 1. Why white-sky and black-sky albedos of each land cover type are similar? Is it due to MODIS which makes observations with a small range of angular configurations? What would be the impact of this approximation considering changes of diffuse radiation in the climate projection?**
- 2. Could you please explain how the projections of future urban land expansion were calibrated with different historical urban land products? It is unclear.**
- 3. The use of green roof is described as an interesting solution. However, large uncertainties exist about the magnitude of their potential impact on other bio-physical processes.**

(To increase the transparency and openness of the reviewing process, the journal do support reviewers signing their reports to authors. Also this review has been made by Dominique Carrer.)

REVIEWER COMMENTS

Reviewer #1 (Remarks to the Author):

Planetary albedo changes caused by future urbanization will warm the earth

Using the instantaneous shortwave radiative forcing at the top-of-the-atmosphere (TOA) as a global climate forcing metric (“RF_TOA”), the study investigates the climate forcing of recent historical, mid-21st century, and end 21st century surface albedo changes associated with urban expansion (or “urbanization”). The authors rely on MODIS land cover and surface albedo products (or their derivatives) along with datasets of projected urbanization and snow cover linked to several SSP-RCP scenarios from a variety of sources. For the most part the work appears thorough and technically sound, and the authors have made sensible use of key datasets to address an important knowledge gap surrounding the climate forcing of urban expansion. Given the chosen climate forcing metric, their conclusions appear sufficiently supported by their analysis. However, the focus solely on surface albedo change is not sufficiently justified, and the chosen metric provides an incomplete and potentially misleading picture of the climate forcing attributable to urban expansion. Land cover changes driven by urban expansion affect surface fluxes of heat and moisture in parallel, resulting in perturbations to atmospheric properties that can lead to additional radiative flux perturbations locally (which can either dampen or enhance the albedo-change induced instantaneous shortwave RF at TOA) as well as trigger remote responses. The comparison of the local albedo change-driven forcing to the global mean forcing from CO₂ emissions is misleading at best and counter-productive at worst.

AN: Thank you for summarizing our work and providing constructive suggestions to improve this manuscript. Firstly, we want to clarify that by using the albedo kernels produced from climate models (even though the future kernels are empirically approximated), we are using stratospherically adjusted radiative forcing (RF), which is distinct from simple instantaneous RF. RF factors the stratospheric temperature adjustment. We agree that land cover changes, including urban expansion, have multiple effects on both the local and global climate systems. Aside from albedo changes, there are changes in carbon fluxes that can produce RF, and changes in water and heat fluxes that may dampen or enhance the albedo- or carbon-induced RF especially locally (though the global effect is very small). We have now added a brief review of albedo-related studies, as well as some justifications for this understudied but important topic (lines 68-78 and 88-102).

We could not agree more with you on that there would be both local response and remote responses from these perturbations. As a global scale study, in the revised manuscript, we focus only on effects on the global mean climate, which may be greatly different from the local/regional changes. The effects on the global mean temperature implicitly suggest that any local changes can aggregate and trigger remote responses. To quantify the global climate change effect, one needs to estimate the top-of-atmosphere, global-scale radiative forcing resulting from a change in surface properties. This is fundamentally different than a change in local surface energy fluxes such as latent and sensible heat fluxes. To explain, local surface evaporative cooling, with its associated latent energy flux, involves a flux of energy into the atmospheric

boundary layer but does not release that energy to outer space. Instead, the latent energy heats the atmosphere when the gaseous water vapor re-condenses. Thus, local surface cooling of evapotranspiration does not cool the global climate system but rather cools only the local area where the evaporation occurred. There can be some second-order effects associated with a change in surface evaporation, involving changes in cloud cover and changes in longwave emission from the upper atmosphere and influencing the global TOA radiative energy budget. However, these effects are relatively subtle. Furthermore, reduced evaporative cooling is often compensated to a certain degree by an increase in sensible heat flux and longwave emission, both of which cool the surface environment. The latter of these has a direct planetary cooling effect for the portion of longwave emission that passes through the atmosphere with emission to outer space, which could be considered in future earth system models. Overall, the surface reflectivity (albedo) that sends incoming shortwave radiation out to space leads to global climate cooling is a primary biophysical quantity that needs to be compared to the global climate effects of carbon emissions for land use land cover changes.

RF, indeed, is not a perfect indicator/metric. We have therefore discussed its limitations (Lines 297-310) and recommended using ERF (per your suggestion later; it is also adopted by IPCC AR6) for future studies since currently we still lack ERF kernels to do such data-driven quantifications. Nevertheless, RF remains widely used and is very useful for understanding the factors driving global mean temperature change. Moreover, IPCC AR5 have suggested, in many cases, ER and ERF are nearly equal or not significantly different from each other with the exceptions of BC-related forcing. Lastly, we are not comparing the local albedo change-driven forcing to the global mean forcing from CO₂ emissions. We are comparing the global forcing from locally aggregated global albedo change to the global mean forcing from CO₂ emissions. Comparing albedo-induced climate effect to CO₂ emissions has been done in many previous publications (R. M. Bright, 2015; R. M. Bright & Lund, 2021; Carrer et al., 2018; Williams et al., 2021). Please see more detailed responses below that are related to your general comments.

The manuscript (especially the title and Introduction paragraphs) seems to be unjustly overstating the significance of the results to make the manuscript Nature-worthy, and many statements need to be revised and appropriately caveated. These caveats and limitations notwithstanding, however, I do think there is value to the empirically based albedo change estimates that can serve as an important reference/benchmark for the climate and UHI modeling community. I will leave it to the Editor to decide if a toned-down interpretation of the results would be of interest to the general audience of Nature Communications.

AN: Thank you for letting us know your first impression of “overstating” while reading our study. We did not do this on purpose but based on our scientific finding. The albedo change caused by future urbanization will indeed produce warming effects to increase the global mean temperature according to our results (please see a more detailed and related response below, where we explain why positive RF equals warming effects). Following your suggestions, nevertheless, we have rewritten the introduction to better single out the significance of studying the climate effect of albedo, revised the caveated statement, and added more limitations in the discussion section. Regarding the title, we also changed to “Albedo Changes Caused by Future Urbanization Contribute to Global Warming” so to more precisely match the revision-quantifying the change in global mean temperature.

Major comments:

Scoping

The authors make no sincere effort to describe the physical processes and mechanisms perturbed at the surface by land cover conversions to urban lands, and how these in turn connect to perturbed atmospheric process driving climate changes across different spatial and temporal scales. Few links to the scientific literature on this front are presented, and thus the singling out of surface albedo change appears not sufficiently well-grounded and justified as a result. Several studies (Zhao et al. 2014; Georgescu et al. 2009a) point to the relevance and importance of surface albedo changes (reference info is provided at the end of this review) in the context of urbanization and local direct temperature effects -- which the authors could draw upon for improved contextualization and justification. On a similar note, no real effort is made to establish the credibility of the chosen RF-based measure as a suitable indicator of the ensuing climate response (i.e., temperature change)

likely to be experienced both locally and remotely. The motivation for moving away from the instantaneous towards “effective” radiative forcing (see IPCC 5AR WG1 Chapter 8; Sherwood et al. 2015; Forster et al. 2016) is that it is a better indicator of the temperature response since the atmospheric adjustments are included (see Georgescu et al. 2009b for an example of rapid atmospheric adjustments in response to surface energy balance changes linked to urban expansion). Further, and on a related note, the authors also make no real effort to discuss how urban expansion might affect climate remote of the urbanized areas, which is important from the regional-to-global adaptation/mitigation perspective. This is also exemplified in Georgescu et al. 2009b. At the global scale, Jacobson & ten Hoeve (your ref. 18) clearly demonstrate that remote responses are larger than the local response and dominate the global mean response to urbanization.

AN: Thank you for these comments and useful references for helping us better define the scope of our study. We have revised our introduction to better describe bio-physical and bio-chemical processes related to land cover changes that can impact climate both locally and regionally/globally (lines 63-66), and to better single out the importance of surface albedo (lines 68-80, 88-104). Regarding the RF metric, we have now established the temperature response to RF using the equilibrium climate sensitivity tools and added a discussion on the limitation of RF and ERF for assessing influences of forcing agents on global mean temperature. Regarding local- and remote- responses, and climate effects of urbanization at different temporal and spatial scales, we want to clarify again that while we agree these are important questions especially in local and regional scale studies, they are not our focus in this study. Our study has a global focus, so we only focus on the effects on the global mean climate (i.e., temperature) that are more meaningful to global climate change. On the other hand, our results regarding positive forcing to the global climate caused by urbanization implicitly suggest that local urbanization can affect global mean temperature and thus the climate of remote areas. In this study, we do not intend to answer questions about where exactly climate effects in un-urbanized areas can be remotely triggered by urbanization in a particular place. Instead, we suggest that these questions are better placed in future work using numerical model/land-atmosphere model at regional scale (Cao et al., 2016; Georgescu et al., 2009a, 2009b). For these reasons and for reducing confusions, in the revision,

we have excluded the analysis on surface radiation changes (i.e., local effect), and only focused on globally aggregated top of atmosphere RF.

A review of the UHI-climate modeling or land use/land cover change climate modeling literature through this lens might help the authors qualitatively define some confidence bounds for their chosen RF measure in terms of its robustness as an indicator climate change, both locally and globally. Caveats and limitations are likely to emerge naturally from a more honest and comprehensive review of the literature in these areas.

AN: Thanks for the suggestions for helping better set the context of our study. In response, we have substantially revised our introduction. We have included important studies in the introduction (thank you for providing some references). For example, we added: “Urban albedo has been shown to affect climate at local to regional and global scales. Zhao et al.²³ found empirical evidence that the night-time urban heat island (UHI) intensity and the urban-rural albedo difference are negatively correlated and argued that increase urban albedo can produce measurable results to mitigate UHI on a large scale. Hu et al.²⁴ assessed the surface albedo change in Beijing from 2001 to 2009 caused by urbanization using remote sensing and field measurements. Their results indicate a positive relationship between albedo-induced radiative forcing and urbanization level, and that the cumulative effects of albedo change caused by urbanization could be important drivers of local climate change”

RF is an indicator for global (mean) climate changes (and as you point out later, it is reserved for quantifying an external energy balance perturbation, thus global effects). Any locally induced RF cannot transfer to a local temperature change. We agree that, under the same global forcing, any local or regional response could differ significantly from the global mean change. Thus, we have clarified in this revision that we are studying the effect of albedo changed on the global mean temperature caused by urbanization. Caveats and limitation are added in the discussion part of the revision, for example:

“RF was widely used and proved to be an effective tool for assessing the influence on global mean temperature of most individual agents affecting Earth’s radiation balance^{45,46}, although it is not a perfect tool. One limitation of RF is its nonadditivity of different RF agents, but this is not an issue in this study as we only studied RF of albedo. Improved understanding has suggested the recently proposed concept of effect RF (ERF) is a better indicator of the temperature response for some forcing agents, because ERF considers other rapid adjustments of troposphere (e.g., atmospheric temperatures, water vapor, and clouds) additional to stratospheric temperature which is the only adjustment for RF. However, the method for calculating ERF remain unsettled, ERFs require more computational resources than RF and spread wider across models due to rapid model-dependent feedbacks, and ERFs can have difficulties of isolating small forcing due to uncertainties relative to the forcing itself. Still, future work should consider adopting to ERF for better estimating albedo-induced radiative forcing when ERF radiative kernels are available.”

UHI is a globally existed phenomenon. While UHI may also contribute to global warming and this global effect may mostly be due to albedo related changes (e.g., Jacobson & ten Hoeve, 2012). UHI captures people’s attention mostly at a local level, as city-level issue that concerns

the temperature difference between urban centers and surrounding rural areas (Steenefeld et al., 2011; Zhou et al., 2014). UHI is more caused by a combination of reduced evapotranspiration and increased solar radiation but change in evapotranspiration has little effect on global mean temperature (IPCC AR5 and 6). This paper is thus not concerned with local UHI. There is one study that focuses on UHI for future urbanization (Huang et al., 2019), which is a global scale study but mostly studies city-scale effects. For this reason, we do not review the abundant literature on UHI, but included most related studies (Jacobson & ten Hoeve, 2012; L. Zhao et al., 2014).

Methods

The appropriateness of the "CO₂-eq." measure notwithstanding, its calculation is poorly described, and the same results are presented in some places with units "Gt-CO₂-eq.," while in others with units "Gt-CO₂-eq./yr". By my estimate, the "per yr" values are two orders of magnitude too high for the reported RF_TOA results. Further, the authors acknowledge a changing future radiation regime as being "important" and "significant" (P7, L250-251), but do not factor this into their analysis. Embedded in the radiative kernels applied to quantify radiative forcing ("RF_TOA") in all future time steps (up to 2100) is the atmospheric state of 2006/2007 (i.e., clouds, aerosols, water vapor, and other factors affecting shortwave radiative transfer), but no justification is given as to their suitability in estimating radiative forcings under future atmospheric conditions. Jonko et al. (2012) for example show substantial deviations (up to ca 20%) in albedo kernels

between those estimated under 8 x CO₂ forcing and those with the present atmospheric background state. The use of the 2006/2007 Pendergrass kernel (ref 36) to estimate forcings in 2050 and 2100 is thus highly questionable. The better approach would be to apply a kernel (or radiative transfer mode) that can account for changes to an evolving atmospheric background state. An example is the model underlying the kernel of Bright & O'Halloran (2019) that appears as a simple function of surface downwelling solar radiation flux – a variable which is freely available in a variety of CMIP5 & 6 archives (alternatively the simplified kernel model of Donohoe et al. (2020)). The authors appear to have relied on such archives for future snow cover, so an extension to included future radiation budget variables as input is not an unreasonable amount of added effort.

AN: These comments are very much appreciated to improve our methodology. First, units of CO₂ equivalence in this study should be all "Gt-CO₂-eq." We have clarified the description of converting albedo-induced RF to CO₂ equivalence (lines 457-466). The conversion of albedo-induced RF to CO₂ equivalence (GWP) is now widely used in literature, and many researchers are seeking to do such conversion as it is very useful in land use forcing research when carbon flux changes accompany the albedo changes (R. M. Bright, 2015; R. M. Bright & Lund, 2021; Carrer et al., 2018). Second, radiation fields affect computation of two quantities: the blue albedo, as it requires proportions of diffuse and direct radiation flux, and the transmittance that is embedded in the kernels and requires surface and top-of-atmosphere total radiation. For the first effect, we have added sensitivity analysis to show that since white- and black-sky albedo do not show large differences, the changes in diffuse/direct radiation ratios have little effect on the final RF (lines 261-279). For the second effect, following your suggestions, we have now used future projected surface and top-of-atmosphere radiation from CMIP5 to empirically approximate kernels for future periods applying Bright & O'Halloran's (2019) method (Lines 393-399).

Lastly, we agree that using the 2006/2007 Pendergrass kernel (Pendergrass et al., 2018) to estimate forcings in 2050 and 2100 is not a very good idea considering the changes of atmosphere conditions (i.e., clouds, aerosols, water vapor, and other factors affecting shortwave radiative transfer). Thanks to your suggestion, we now use an empirical kernel based on future projection of surface and top-of-atmosphere shortwave radiation. It is not perfect but is the best available tool. The transmittance based on surface/TOA irradiance ratio can reflect the effects of aerosols, cloud, and water vapor.

Other general comments

The English grammar needs improvement.

AN: we have used professional language editor to improve the grammar.

Other specific comments

P1, Title: I suggest an alternate title without the word “warm” that is more faithful to the content and scoping and which is more indicative of the actual metric applied in the work (e.g., instantaneous SW RF).

AN: We slightly changed the title following your comments but did not exclude the word “warm”. However, to better reflect the title, we have revised the text to link RF to global mean temperature changes. We excluded the word planetary to avoid confusion between planetary and surface albedo which are different in some settings, as we are using surface albedo. The new title is “Albedo Changes Caused by Future Urbanization Contribute to Global Warming”.

P2, L45, Abstract: Please refrain from referring to positive and negative RF results as “warming” and “cooling” here and elsewhere throughout the manuscript. “Warming” and “cooling” imply that some temperature indicator has been investigated, which has not.

AN: We have now directly linked RF to global mean temperature changes. However, even without converting to global mean temperature change, we argue we can still use “warming” or “cooling” interchangeably with positive or negative radiative forcing. RF, is indeed designed an indicator of global temperature changes, as RF can directly change into global mean temperature response:

$$\Delta T_s / RF = \lambda$$

Where ΔT_s is the global mean surface temperature, and λ is the climate sensitivity parameter. While λ has uncertainties and vary among models, it is always a positive number. Other unconsidered feedback/adjustment may dampen or amplifying λ , but no current knowledge has shown it can reverse the sign of λ . In fact, IPCC AR5 (which still mostly using RF not ERF) clearly state that:

“Radiative forcings greater than zero lead to a near-surface warming, and radiative forcings less than zero lead to a cooling.”

IPCC AR5 also uses them the same way as we used them in our study. We quote some examples from IPCC AR5:

“The total anthropogenic radiative forcing over 1750–2011 is calculated to be a warming effect of 2.3 [1.1 to 3.3] W/m² (Figure 1.4)”

“The radiative forcing from stratospheric volcanic aerosols can have a large cooling effect on the climate system for some years after major volcanic eruptions”.

P3, L55: “Land cover and land use change (LCLUC)”. Please consider aligning with the prevailing acronym of the scientific literature: Land use/land cover change (LULCC).

AN: Changed as suggested. Thank you.

P3, L 57-58: Possible to indicate what the current global urban area occupation is? It would help the reader acquire an immediate feel for the scale of the research problem being addressed.

AN: Yes, we have added that (line 57): The global urban land is currently 0.79 million km² according to MODIS LULC in 2018.

P3, L58-60: I think it is enough to just say “physical properties” here instead of the long and seemingly arbitrary list of examples.

AN: Changed as suggested. Thank you.

P3, L63-64: The preceding sentence mentions several climate forcing pathways or mechanisms, so it is unclear why there is a sudden shift here towards the focus on albedo. A justification for this shift in focus is needed, since the likelihood of a reader not having a background in the subject matter is higher for a more generalist journal like Nat. Comm. In fact, the merit of this entire study hinges on the relevance of surface albedo, and thus it needs to be sufficiently demonstrated that surface albedo changes in fact have high relevance. See the first major comment above surrounding the article scoping/framing.

AN: we have deleted this sentence since the transition is not smooth. We have addressed the relevance of surface albedo changes in other places in the introduction:

“Anthropogenic LULCC, including urbanization which is defined as the expansion of urban land here, affect local-to-global climate through not only modifications in surface roughness, carbon, and latent heat flux, but also changes in surface albedo that directly altering Earth radiation budget⁹. However, most of the attention has focused on quantifying changes in carbon processes^{10–14}, and less attention is paid to albedo changes^{9,15–18},”

“Urban albedo has been shown to affect climate at local to regional and global scales. Zhao et al²³ found empirical evidence that the night-time urban heat island (UHI) intensity and the urban-rural albedo difference are negatively correlated and argued that increase urban albedo can produce measurable results to mitigate UHI on a large scale. Hu et al.²⁴ assessed the surface albedo change in Beijing from 2001 to 2009 caused by urbanization using remote sensing and field measurements. Their results indicate a positive relationship between albedo-induced radiative forcing and urbanization level, and that the cumulative effects of albedo change caused by urbanization could be important drivers of local climate change.”

P3, L65-68: This sentence needs re-writing to improve clarity and technicality. How does

“albedo, evapotranspiration, and heat conduction” “affect local-to-global climate” as measured with the metric “radiative forcing (RF)”? Surface “albedo” is a state property, but RF is a measure of some state change (i.e., albedo change). Similarly, “evaporation” and “heat conduction” are fluxes characterizing the state of the system, and changes to these do not directly perturb the planetary energy balance (i.e., do not contribute to RF).

AN: we have rephrased this sentence to improve clarity (lines 63-65). Yes, evaporation and heat conduction are fluxes that do not directly perturb the planetary energy balance. RF in this study was only used to describe the perturbation to the planetary energy balance caused by changes in albedo.

P3, L68: Please provide more details about this important reference in the reference list which appears only as “IPCC Climate Change 2014”. I assume the authors are referring here to the 5th Assessment Report – specifically the work of Working Group I – and specifically the radiative forcing chapter (Chapter 8, Myhre et al. 2013)?

AN: More details on these references has been provided (Reference No. 9) Yes, it is Chapter 8 from to the 5th Assessment Report – specifically the work of Working Group I. Thank you.

P3, L73: Replace “cooling effect” with “negative radiative forcing” ...also L76 and elsewhere.

AN: please see our response to your comments on “P2, L45”. Nevertheless, we have changed this and many other sentences as suggested when it is not necessary to explicitly state cooling or warming.

P4, Section 2: Consider re-naming this section since the main results appear to be presented in the previous (and last) sentence of Section 1. Alternately consider presenting the results of L122-124 somewhere in Section 2 instead.

AN: Done as suggested. We have moved L122-124 to Section 2.

P4, L148: I suggest using alternative terminology and acronym to “surface radiative forcing” for surface level radiative flux perturbations, as “radiative forcing” is conventionally reserved for an external (i.e., planetary) energy balance perturbation. How about “surface shortwave forcing” with acronym “DeltaSW_SUF” or something.

AN: We appreciate your suggestion of using “surface shortwave forcing” and agree that “radiative forcing” is conventionally reserved for an external energy balance perturbation, although we adopted it from a previous study that used “surface radiative forcing” to describe perturbations to Earth’s surface energy balance (Feldman et al., 2015). Nevertheless, in this revision, we have excluded the results regarding to surface shortwave forcing and instead added results of surface albedo changes directly that is more relevant to the final goal, i.e., quantifying the global top of atmosphere radiative forcing that may drive changes of the climate. In this study, we are not concerned about the local changes in surface shortwave radiation. We also changed RF_{TOA} to RF for simplicity, as RF is usually reserved for top-of-atmosphere perturbation already.

P4, L128: What is meant by surface radiative forcing “changes”? RF denotes a change (or difference) in radiative flux with respect to some reference state. Is the reference albedo state

here the flux associated with 2001 albedo, i.e., $RF = f(\text{alb}_{2018} - \text{alb}_{2001})$, in which case the word “change” is not needed? Or has an RF been defined also for 2001 (e.g., albedo change from pre-industrial?), and if so, what is the reference year that is used?

AN: Yes, RF denotes a change, and the “word” change is not needed. The reference albedo here is year 2001, so $RF = f(\text{alb}_{2018} - \text{alb}_{2001})$ is right. However, since we do not present results of surface radiative forcing in the revision, this part is deleted.

P14, Figure 2: It is impossible to see differences between the right and left column of results (i.e., 2050-2018 vs. 2100-2028 results). It would be more informative if the right column instead showed the 2100 differences with respect to the 2050 results. As for Figure 1, consider also going with a different map projection that maximizes the area(s) of interest, i.e., the land areas between -55°S and 70°N. Or consider cropping out unnecessary latitude and longitude bands.

AN: We appreciate the suggestion. We now show the differences in 2100 with respect to the 2050 for both figure 1 and 2, and changed the projections to Robinson to better show land areas between -55°S and 70°N.

P5, L138: You mean local forcing?

AN: Yes, I mean local surface forcing. However, in the revision, we have excluded patterns of local surface forcing, and focused on top of atmosphere RF that can affect global climate.

P5, L164: “albedo-induced warming effect”. Not to beat a dead horse, but again this terminology is incorrect and misleading, as the metric under scope is the (instantaneous shortwave) RF at TOA (in W/m^2), and it is unknown what the temperature response at the surface is, which might be opposite to the instantaneous SW RF at TOA, particularly areas remote of where the urbanization occurs (see your refs. 4 & 18 and Section 2.4).

AN: Please see more details in our response to your comments “P2, L45”. The RF we estimated is the global total RF and, yes, we agree that locally or regionally, the response of local/regional surface temperature may be very different from or opposite to RF, but the global mean surface temperature would have the same sign as RF, i.e., positive RF leads to global warming, and negative RF leads to global cooling. We also have estimated the temperature response at the surface (global mean) in this revision, with the uncertainties in temperature sensitivities and in radiative forcing both considered.

P6, L206-210: The CO₂-eq. results here (and in the Abstract and on P3, L122-124) are given with units “per year”, but they appear similar in magnitude to those presented on P5, L167-173 and in Figure 5 for the same magnitude RF which are NOT given on a “per year” basis. Please explain the discrepancy (and see my comment below on Section 4.4).

AN: The CO₂-eq. are not with a unit of “per year”, it should be simply emission at $t=0$ over a 100-year horizontal periods. We have corrected this mistake.

P7, Section 2.4: The authors mention that the state of the present-day atmosphere – specifically the monthly shares of diffuse and direct solar radiation - is extrapolated to the future when applied to estimate future albedo changes associated with future urbanization. They acknowledge that this “may change significantly” in the future, but yet that this would not affect their future albedo calculations since the direct and diffuse albedos are “similar”. The authors however do not discuss the implications that this same assumption has on their future RF_{TOA} estimates,

calculated here using a radiative kernel based on a 2006/2007 atmospheric state. Albedo kernels are sensitive to clouds, aerosols, and water vapor (affecting shares of direct vs. diffuse radiation), so the authors should either demonstrate that the consequence of this decision has little impact on the results or consider modifying their approach.

AN: Thanks, this is an important point. We have now added the sensitivity analysis to show that changing direct/diffuse radiation has little effect on the final RF computation. Firstly, we compute RF in projected future (2050 and 2100 relative to 2001) based on inversed direct/diffuse radiation ratios from NCEP data and compared it to the original direct/diffuse radiation ratios. Secondly, in the ensemble CMIP5 model we used, three models provide surface diffuse radiation. We used the diffuse radiation, the total surface radiation, and the solar zenith angle, and computed the 3-model mean of direct/diffuse radiation ratios for each month. We then computed the RF based on these direct/diffuse ratios and compared it to the original computation based on NCEP data. The first experiment represents a more extreme condition where the monthly diffuse radiation is large than direct radiation, a rare situation. The first experiment produces an <9% change in RF in all illustrative scenarios (SSP1-2.6, SSP2-4.5, and SSP5-8.5). The second experiment represent a more reasonable change of direct/diffuse radiation, which produces a <4% changes in all three illustrative scenarios and is smaller than the uncertainties in RF caused by uncertainties in urbanization simulations (i.e., from 100 simulation for each SSP scenario)

P8, L280: Are the authors sure that these SSPs are compatible with all three RCPs investigated? See for example Kebede et al. (2018). It is not made clear in the methods which SSPs are combined with the three RCPs, although an inspection of Figure 5 suggests that compatibility considerations have been ignored.

AN: You are raising an important point regarding the compatibility between SSPs and RCPs. We indeed considered this before but did not find a consistent standard. In this revision, we adopted three illustrative scenarios based on IPCC AR6 in the main contents, i.e., SSP1-2.6, SSP2-4.5, and SSP5-8.5, which are corresponding to low, medium, and very high emission scenarios. In the discussion part where we have included more scenarios to discuss the uncertainties caused by different projections of urbanization, we have adopted the compatibility matrix from Riahi et al. (2017).

P9, L 313: Why this older v005 product and not the latest MODIS snow cover product (v006) whose accuracy has been demonstrably and notably improved, notably due to improvements in the snow detection algorithm? (see Masson et al. 2018). Also, what variable from the MOD10CM product is being used? I assumed fractional snow cover (FSC) upon first reading but am now in doubt when looking at Eqs. 2 & 3 below which seem not to make use of FSC but some binary flag instead (either 1 for snow-free or 0 for snow), in which case this should be stated clearly. If FSC is in fact used in Eqs. 2 & 3, then these equations appear incorrectly formulated, as FSC should be used to weight the monthly snow-free and snow-covered albedos sourced from the look-up maps.

AN: We indeed used the latest version 6.1 (DOI:10.5067/MODIS/MOD10CM.061, please note the data set ID is MOD10CM). We now have specified the version to be clear about that in the data description. It is the fractional snow cover used from MOD10CM, as explained following equation 3. The equation is a weighted sum, i.e., the binary flag ($s=0$ or 1) is used to represent

the fraction of snow free ($f_{s=0}$) and snow cover free ($f_{s=1}$). We have reconstructed the equations to consider this (new equations 2, 3 and 5).

P9, L320: “biased in MODIS observations”? Please correct.

AN: Thanks for noting this. It is corrected in the revised version of the manuscript.

P11, Section 4.4: The units provided here for Eq. (9) are “Gt-CO₂-eq.” as well as those for the results presented in Figure 5. But the same results appear to be presented in the Abstract as “Gt-CO₂-eq./yr” which is a two-orders-of-magnitude discrepancy when summed over TH = 100 years. What is being integrated to TH in the numerator of Eq. (9)? The relationship between variable “RF_delta_alb” and time (“t”) is unclear. Are the transient (or time-dependent) RF_d_alb pathways presented in Figure 3 being used here? Or does “RF_d_alb” correspond to the albedo differences between 2050 (or 2100) and 2018 – and if so – is this value being multiplied by TH? Please improve the description of the CO₂-eq. calculation methodology and harmonize variable naming to be consistent across equations. An annual time step is absent from all “RF” and “d_alb” equations of sections 4.1 and 4.2.

AN: “Gt-CO₂-eq./yr ” is a mistake. Thanks for catching this. We corrected it and made it consistent across the texts. What is being integrated to TH in the numerator is the RF caused by albedo changes, i.e., $RF_{\Delta\alpha}(t)$, because albedo-induced RF does not decay, the numerator equals to $TH * RF_{\Delta\alpha}(t=0)$. The $RF_{\Delta\alpha}$ corresponds to the RF caused by albedo differences between 2018 and 2001 for the past, and between 2050 and 2018, and 2100 and 2018 for the future. And, yes, this value is multiplied by TH. Note we previously used rfCO₂ (global total forced radiation) which is roughly the currently used kCO₂ (radiative efficiency) multiplied by the global area. Since our estimation is at global scale, the total global area in both the numerator and denominator can be canceled, we therefore updated the equation to use this simplified equation following Bright et al., (2015). We also added annual time steps to the equations.

P13, Figure 1. Differences between panels (c) and (b) are imperceptible. Please consider showing instead the difference from 2100-2050 as panel (c) so that the additional change becomes apparent.

AN: Done as suggested.

P14, Figure 2. Similar comment as above – differences between panels (b) and (a) are imperceptible. The same goes for the differences between panels (d) and (c). Please consider an alternate way of presenting the differences between 2100 and 2050.

AN: Done as suggested.

References

Jonko et al 2012. Climate Feedbacks in CCSM3 under Changing CO₂ Forcing. Part I: Adapting the Linear Radiative Kernel Technique to Feedback Calculations for a Broad Range of Forcings, *Journal of Climate*, 25, 5260-5272, 10.1175/jcli-d-11-00524.1.

Zhao et al. 2014. Strong contributions of local background climate to urban heat islands, *Nature*, 511, 216-219, 10.1038/nature13462.

Georgescu et al 2009a. Climatic effects of 30 years of landscape change over the Greater Phoenix, Arizona, region: 1. Surface energy budget changes, Journal of Geophysical Research: Atmospheres, 114, <https://doi.org/10.1029/2008JD010745>.

Georgescu et al. 2009b. Climatic effects of 30 years of landscape change over the Greater Phoenix, Arizona, region: 2. Dynamical and thermodynamical response, Journal of Geophysical Research: Atmospheres, 114, <https://doi.org/10.1029/2008JD010762>.

Sherwood et al. 2015. Adjustments in the Forcing-Feedback Framework for Understanding Climate Change, Bulletin of the American Meteorological Society, 96, 217-228, 10.1175/bams-d-13-00167.1.

Forster et al. 2016. Recommendations for diagnosing effective radiative forcing from climate models for CMIP6, Journal of Geophysical Research: Atmospheres, 121, 412,460-412,475, 10.1002/2016jd025320.

Bright & O'Halloran 2019. Developing a monthly radiative kernel for surface albedo change from satellite climatologies of Earth's shortwave radiation budget: CACK v1.0, Geosci. Model Dev., 12, 3975-3990, 10.5194/gmd-12-3975-2019.

Donohoe et al. 2020. The Effect of Atmospheric Transmissivity on Model and Observational Estimates of the Sea Ice Albedo Feedback, Journal of Climate, 33, 5743-5765, 10.1175/jcli-d-19-0674.1.

Masson et al 2018. An Assessment of Existing Methodologies to Retrieve Snow Cover Fraction from MODIS Data, Remote Sensing, 10, 619.

Kebede et al 2018. Applying the global RCP–SSP–SPA scenario framework at sub-national scale: A multi-scale and participatory scenario approach, Science of The Total Environment, 635, 659-672, <https://doi.org/10.1016/j.scitotenv.2018.03.368>.

AN: Thanks for providing these references. They are very useful and have been incorporated.

Reviewer #2 (Remarks to the Author):

Major comments

Radiative kernels are here used to compute TOA RF due to albedo changes. The authors use CESM1.1.2 tool to make the calculation. The authors do not explain if this tool also take into account the associated changes in evapotranspiration. I understand that it is not the case. In my opinion, the study has to show the impact on evapotranspiration and potentially evaluate the uncertainties due to cloudiness changes. Moreover, the authors should also add a life cycle analysis (i.e., to build new urban areas potentially generates important greenhouse gas emissions).

The study does not discuss the impact on carbon fluxes - to remove vegetation areas potentially leads to important changes of carbon sequestration by the surface.

AN: We agree both evapotranspiration and carbon flux are very important processes related to urbanization (or other land use changes) that can produce climate impacts.

The kernel does not directly consider the associated changes in evapotranspiration due to the studying agent (i.e., urbanization). However, the secondary effect of evapotranspiration (water vapor, cloudless) on light transmittance is considered in the kernels (both the CESM and the new empirically approximated kernels for future projection quantifying transmittance). The direct climate effects of changes in evapotranspiration are not included in our study. There are many studies suggesting that the non-radiative process evapotranspiration can have a major influence on local/regional surface temperatures during urbanization (e.g., reduced evapotranspiration caused temperature increase in urbanized area, and thus UHI) (Fitria et al., 2019; Mazrooei et al., 2021; G. Zhao et al., 2019). However, the approach used in those studies, of analyzing land surface temperature and the surface energy balance, does not quantify the global-scale climate change impact that can be equated to the effect of greenhouse gases in the atmosphere. To quantify the planetary climate change effect, one needs to estimate the top-of-atmosphere, global-scale radiative forcing that results from a change in surface properties. This is fundamentally different from a change in local surface energy fluxes such as latent and sensible heat fluxes. To explain, local surface evaporative cooling, with its associated latent energy flux, involves a flux of energy into the atmospheric boundary layer but does not release that energy to outer space. Instead, the latent energy heats the atmosphere when the gaseous water vapor re-condenses. Thus, local surface cooling of evapotranspiration does not cool the climate system but rather cools only the local area where the evaporation occurred. There can be some second-order effects associated with a change in surface evaporation, involving changes in cloud cover and changes in longwave emission from the upper atmosphere and influencing the global TOA radiative energy budget however these effects are relatively subtle. Furthermore, reduced evaporative cooling is often compensated to a certain degree by an increase in both sensible heat flux and longwave emission, both of which cool the surface environment. The latter of these has a direct planetary cooling effect for the portion of longwave emission that passes through the atmosphere with emission to outer space, which could be considered in future earth system models. Overall, surface reflectivity (albedo) that sends incoming shortwave radiation out to space and leading to global climate cooling is a primary biophysical quantity that needs to be compared to the global climate effects of carbon emissions for land use land cover changes. That is why, IPCC reports states that scientists consider tropospheric water vapor a feedback agent, rather than a forcing to climate change.

Land use land cover changes do greatly impact carbon cycling and contribute to anthropogenic carbon emissions (Hong et al., 2021; Houghton et al., 2012; Houghton & Nassikas, 2017). The impact of past urbanization on carbon cycling, has been partly covered by previous studies. For example, Liu et al. (2019) studied the impact of global urbanization on terrestrial net primary productivity (NPP) from 2000 to 2010; Seto et al. (2012) studied the loss of biomass with projected global urbanization from 2000 to 2030; and Vasenev et al. (2018) studied the changes in soil carbon stock in the Moscow region of projected urbanization in different scenarios. To study the net carbon changes, one must comprehensively and simultaneously study the changes

in carbon flux (NPP) and stocks (above/below ground biomass, and soil carbon). By using a dynamic land ecosystem model, Zhang et al. (2012) was able to do an overall estimation for C balance in urban ecosystems in the southern United States. However, no study has so far been able to do this globally due to the complexity and lack of observational data. As you have suggested, to study the carbon flux/pool changes, a life cycle analysis is needed, since the carbon dynamic is different at different stages. For example, Zhang et al. (2012) show that after a rapid decline of carbon storage during land conversion, urban land can gradually accumulate carbon and may compensate for the initial carbon loss in 70–100 years. A life cycle analysis is also needed because the related carbon processes are greatly impacted by climate change, i.e., CO₂ concentration, temperature, and water availabilities. Currently, we still do not have a clear mechanistic understanding of how NEP, soil respiration, and below/above ground biomass response to future increased temperature and changed water availabilities. It is thus out of scope of this study to do a comprehensive study on the net carbon changes of projected future urbanization using data-driven approaches. In this study, we wish to solve the single problem of climate impacts of albedo changes due to urbanization which, to our knowledge, had not addressed before. Future studies may apply high resolution dynamic vegetation/land ecosystem models or other earth system models to do a comprehensive estimate of the net impacts of urbanization in a life cycle of 50-100 years. While we could not do such a comprehensive analysis of carbon consequences, we have added more discussion about it.

It is not clear how snow covers the urban area. Is there a fraction of snow cover with a snow depth?

Finally, it seems that the authors have used a stable solar radiation. They argue that the effect of solar radiation variation on their estimates are limited, as the change of total solar radiation is slow. Please could the authors show solar radiation changes in the different climate projections? Could they show the change also between the direct and diffuse components? And could they discuss the impact of an albedo increase on the increase of multi-scattering contribution bounced back to the surface.

AN: It is the fraction of time (in each month) during which the land (including urban area and other land covers) is covered by snow. Snow depth is not used or included in our data. For the assumption of stable solar radiation, we no longer make this assumption in the revision. An empirical kernel has been computed using projected total solar radiation and surface radiation. The transmissivity as defined by the ratio of surface radiation to the top of atmosphere radiation was used in computing the kernel (see Equation 1), which can partly consider the effect of clouds. Clouds will also change the direct and diffuse components, and thus our computation of blue albedo. Due to the lack of direct and diffuse radiation components in CMIP5 (also in CMIP6), we have assumed constant direct and diffuse radiation composition based on historical data. To estimate the uncertainty effect of this, we have added a sensitivity analysis to show the limited effect of varying direct/diffuse radiation ratio on our RF estimates. Firstly, we compute RF under the extreme condition, where the diffuse/direct radiation ratio was inversed from NCEP data, making monthly diffusing radiation larger than direct radiation, a rare observed situation. Our result suggests a change of RF of less than 9%. Second, in the ensemble CMIP5 model we used, there are three models that provide surface diffuse radiation. We used the diffuse radiation, the total surface radiation, and the solar zenith angle, and computed the 3-model mean of diffuse/direct radiation ratio in each month. We computed the RF based on these direct/diffuse

ratios and compared to the original computation based on NCEP data. This experiment with a more reasonable change of direct and diffuse radiation components shows very little effect on the final RF, a change of <4% in all illustrative scenarios (Appendix B). Lastly, we understand that increased albedo can affect downwelling radiation because of multi-scattering, which can further influence diffuse/direct radiation ratio and the surface/top of atmosphere radiation ratio. Such effects have been largely embedded in the GCM-derived kernels that fully consider atmospheric transmission. It is also implicitly in the empirical kernel method of Bright et al (2019) which we adopted for RF estimate of future urbanization. Moreover, the MODIS albedo product also has taken into multi-scattering as part of atmosphere transmission as it is observed from the space rather than from the ground level.

To conclude, the study is very interesting, and conclusions are original. However, the potential impacts on other bio-physical processes are not enough analyzed and discussed (in my opinion).

AN: Thank you. Your major comments have helped us better frame the study. And we have added more discussions on the potential impacts of other bio-physical processes.

Minor comments

1. Why white-sky and black-sky albedos of each land cover type are similar? Is it due to MODIS which makes observations with a small range of angular configurations? What would be the impact of this approximation considering changes of diffuse radiation in the climate projection?

AN: It is not exactly known why white-sky and black-sky albedos of each land cover type are similar, but we made this statement based on MODIS observations. Your hypothesis might be one of the reasons, as MODIS observations are usually taken at certain times during a day thus limited angular configuration though BRDF model has been applied using all good data measured over 16-day periods. A second reason might be that we used monthly average albedos, that further smoothed out the difference between white-sky and black-sky albedo (e.g., outliers due to intense weather events being smoothed out). Lastly, albedo for most land covers (except for snow and ice) has small values, so their differences are even smaller. The change of diffuse radiation in future climate will impact blue albedo and finally the RF. However, our sensitivity analysis suggests this would be small, most likely <5%, which is smaller than the uncertainties caused urbanization simulations.

2. Could you please explain how the projections of future urban land expansion were calibrated with different historical urban land products? It is unclear.

AN: The future urban land projection was calibrated using the historical urban land areas for the years 1975, 1990, 2000 and 2014, which are acquired from the Global Human Settlement Layer (GHSL) dataset (Martino et al., 2016). The agreement between the simulated and observed urban land expansion is evaluated using the FoM (Figure of Merit) indicator (Pontius et al., 2008). FoM was used because it avoids the drawback of accuracy overestimation in conventional validation metrics (e.g., the Kappa coefficient) (Chen et al., 2020). We have added this information in the data description section of the main text regarding future urbanization projections.

3. The use of green roof is described as an interesting solution. However, large uncertainties exist about the magnitude of their potential impact on other bio-physical processes.

AN: Good point. Thank you. We have added a discussion (lines 236-242) about considering the possible impacts of green roofs on other bio-physical processes when applying green roofs.

(To increase the transparency and openness of the reviewing process, the journal do support reviewers signing their reports to authors. Also this review has been made by Dominique Carrer.)

REFERENCES:

1. Bright, R. M. (2015). Metrics for biogeophysical climate forcings from land use and land cover changes and their inclusion in life cycle assessment: A critical review. In *Environmental Science and Technology*. <https://doi.org/10.1021/es505465t>
2. Bright, R. M., & Lund, M. T. (2021). CO₂-equivalence metrics for surface albedo change based on the radiative forcing concept: A critical review. In *Atmospheric Chemistry and Physics* (Vol. 21, Issue 12). <https://doi.org/10.5194/acp-21-9887-2021>
3. Bright, R., & O'Halloran, T. (2019). Developing a monthly radiative kernel for surface albedo change from satellite climatologies of Earth's shortwave radiation budget: CACK v1.0. *Geoscientific Model Development*, 12(9). <https://doi.org/10.5194/gmd-12-3975-2019>
4. Cao, Q., Yu, D., Georgescu, M., & Wu, J. (2016). Impacts of urbanization on summer climate in China: An assessment with coupled land-atmospheric modeling. *Journal of Geophysical Research*, 121(18). <https://doi.org/10.1002/2016JD025210>
5. Carrer, D., Pique, G., Ferlicoq, M., Ceamanos, X., & Ceschia, E. (2018). What is the potential of cropland albedo management in the fight against global warming? A case study based on the use of cover crops. *Environmental Research Letters*. <https://doi.org/10.1088/1748-9326/aab650>
6. Chen, G., Li, X., Liu, X., Chen, Y., Liang, X., Leng, J., Xu, X., Liao, W., Qiu, Y., Wu, Q., & Huang, K. (2020). Global projections of future urban land expansion under shared socioeconomic pathways. *Nature Communications*, 11(1), 537. <https://doi.org/10.1038/s41467-020-14386-x>
7. Feldman, D. R., Collins, W. D., Gero, P. J., Torn, M. S., Mlawer, E. J., & Shippert, T. R. (2015). Observational determination of surface radiative forcing by CO₂ from 2000 to 2010. *Nature*, 519(7543). <https://doi.org/10.1038/nature14240>
8. Fitria, R., Kim, D., Baik, J., & Choi, M. (2019). Impact of Biophysical Mechanisms on Urban Heat Island Associated with Climate Variation and Urban Morphology. *Scientific Reports*, 9(1). <https://doi.org/10.1038/s41598-019-55847-8>
9. Georgescu, M., Miguez-Macho, G., Steyaert, L. T., & Weaver, C. P. (2009a). Climatic effects of 30 years of landscape change over the Greater Phoenix, Arizona, region: 1. Surface energy budget changes. *Journal of Geophysical Research Atmospheres*, 114(5). <https://doi.org/10.1029/2008JD010745>
10. Georgescu, M., Miguez-Macho, G., Steyaert, L. T., & Weaver, C. P. (2009b). Climatic effects of 30 years of landscape change over the Greater Phoenix, Arizona, region: 2.

- Dynamical and thermodynamical response. *Journal of Geophysical Research*, 114(D5).
<https://doi.org/10.1029/2008jd010762>
11. Hong, C., Burney, J. A., Pongratz, J., Nabel, J. E. M. S., Mueller, N. D., Jackson, R. B., & Davis, S. J. (2021). Global and regional drivers of land-use emissions in 1961–2017. *Nature*, 589(7843). <https://doi.org/10.1038/s41586-020-03138-y>
 12. Houghton, R. A., House, J. I., Pongratz, J., van der Werf, G. R., Defries, R. S., Hansen, M. C., le Quéré, C., & Ramankutty, N. (2012). Carbon emissions from land use and land-cover change. *Biogeosciences*, 9(12). <https://doi.org/10.5194/bg-9-5125-2012>
 13. Houghton, R. A., & Nassikas, A. A. (2017). Global and regional fluxes of carbon from land use and land cover change 1850–2015. *Global Biogeochemical Cycles*, 31(3). <https://doi.org/10.1002/2016GB005546>
 14. Huang, K., Li, X., Liu, X., & Seto, K. C. (2019). Projecting global urban land expansion and heat island intensification through 2050. *Environmental Research Letters*, 14(11). <https://doi.org/10.1088/1748-9326/ab4b71>
 15. Jacobson, M. Z., & ten Hoeve, J. E. (2012). Effects of urban surfaces and white roofs on global and regional climate. *Journal of Climate*. <https://doi.org/10.1175/JCLI-D-11-00032.1>
 16. Liu, X., Pei, F., Wen, Y., Li, X., Wang, S., Wu, C., Cai, Y., Wu, J., Chen, J., Feng, K., Liu, J., Hubacek, K., Davis, S. J., Yuan, W., Yu, L., & Liu, Z. (2019). Global urban expansion offsets climate-driven increases in terrestrial net primary productivity. *Nature Communications*, 10(1), 5558. <https://doi.org/10.1038/s41467-019-13462-1>
 17. Mazrooei, A., Reitz, M., Wang, D., & Sankarasubramanian, A. (2021). Urbanization Impacts on Evapotranspiration Across Various Spatio-Temporal Scales. *Earth's Future*, 9(8). <https://doi.org/10.1029/2021ef002045>
 18. Pendergrass, A. G., Conley, A., & Vitt, F. M. (2018). Surface and top-of-Atmosphere radiative feedback kernels for cesm-cam5. *Earth System Science Data*. <https://doi.org/10.5194/essd-10-317-2018>
 19. Pesaresi Martino; Ehrlich Daniele; Ferri Stefano; Florczyk Aneta; Carneiro Freire Sergio Manuel; Halkia Stamatia; Julea Andreea Maria; Kemper Thomas; Soille Pierre; Syrris Vasileios. (2016). Operating procedure for the production of the Global Human Settlement Layer from Landsat data of the epochs 1975, 1990, 2000, and 2014.
 20. Pontius, R. G., Boersma, W., Castella, J. C., Clarke, K., Nijs, T., Dietzel, C., Duan, Z., Fotsing, E., Goldstein, N., Kok, K., Koomen, E., Lippitt, C. D., McConnell, W., Mohd Sood, A., Pijanowski, B., Pithadia, S., Sweeney, S., Trung, T. N., Veldkamp, A. T., & Verburg, P. H. (2008). Comparing the input, output, and validation maps for several models of land change. *Annals of Regional Science*, 42(1). <https://doi.org/10.1007/s00168-007-0138-2>
 21. Riahi, K., van Vuuren, D. P., Kriegler, E., Edmonds, J., O'Neill, B. C., Fujimori, S., Bauer, N., Calvin, K., Dellink, R., Fricko, O., Lutz, W., Popp, A., Cuaresma, J. C., KC, S., Leimbach, M., Jiang, L., Kram, T., Rao, S., Emmerling, J., ... Tavoni, M. (2017). The Shared Socioeconomic Pathways and their energy, land use, and greenhouse gas emissions implications: An overview. *Global Environmental Change*. <https://doi.org/10.1016/j.gloenvcha.2016.05.009>
 22. Seto, K. C., Güneralp, B., & Hutyra, L. R. (2012). Global forecasts of urban expansion to 2030 and direct impacts on biodiversity and carbon pools. *Proceedings of the National Academy of Sciences of the United States of America*. <https://doi.org/10.1073/pnas.1211658109>

23. Steeneveld, G. J., Koopmans, S., Heusinkveld, B. G., van Hove, L. W. A., & Holtslag, A. A. M. (2011). Quantifying urban heat island effects and human comfort for cities of variable size and urban morphology in the Netherlands. *Journal of Geophysical Research Atmospheres*, 116(20). <https://doi.org/10.1029/2011JD015988>
24. Vasenev, V. I., Stoorvogel, J. J., Leemans, R., Valentini, R., & Hajiaghayeva, R. A. (2018). Projection of urban expansion and related changes in soil carbon stocks in the Moscow Region. *Journal of Cleaner Production*, 170. <https://doi.org/10.1016/j.jclepro.2017.09.161>
25. Williams, C. A., Gu, H., & Jiao, T. (2021). Climate impacts of U.S. forest loss span net warming to net cooling. *Science Advances*, 7(7). <https://doi.org/10.1126/sciadv.aax8859>
26. Zhang, C., Tian, H., Chen, G., Chappelka, A., Xu, X., Ren, W., Hui, D., Liu, M., Lu, C., Pan, S., & Lockaby, G. (2012). Impacts of urbanization on carbon balance in terrestrial ecosystems of the Southern United States. *Environmental Pollution*, 164. <https://doi.org/10.1016/j.envpol.2012.01.020>
27. Zhao, G., Dong, J., Cui, Y., Liu, J., Zhai, J., He, T., Zhou, Y., & Xiao, X. (2019). Evapotranspiration-dominated biogeophysical warming effect of urbanization in the Beijing-Tianjin-Hebei region, China. *Climate Dynamics*, 52(1–2). <https://doi.org/10.1007/s00382-018-4189-0>
28. Zhao, L., Lee, X., Smith, R. B., & Oleson, K. (2014). Strong contributions of local background climate to urban heat islands. *Nature*. <https://doi.org/10.1038/nature13462>
29. Zhou, D., Zhao, S., Liu, S., Zhang, L., & Zhu, C. (2014). Surface urban heat island in China's 32 major cities: Spatial patterns and drivers. *Remote Sensing of Environment*, 152. <https://doi.org/10.1016/j.rse.2014.05.017>

REVIEWER COMMENTS

Reviewer #3 (Remarks to the Author):

As requested by the editor, I've focused on the points raised by the first reviewer. It seems to me that the authors have carefully addressed the issues raised by the first reviewer. Remarks are taken seriously and suggestions have been followed.

Reviewer #4 (Remarks to the Author):

This paper addresses the issue of what impact will increasing urbanisation have on one specific land surface parameter which will control urban heat islands, namely albedo. They employ a LULC dataset of unknown provenance and accuracy and then project forward using different GHG scenarios based on current forecasts towards 2050 and the end of the century. They do not test their model forecast using hindcasting.

Recently, Guo et al. (2022 online from mid December 2021) published an analysis of 10 megacity environments within China which indicated that Radiative Forcing from these cities was POSITIVE when using fine resolution Landsat imagery over a 40 year time-scale and that these results were the opposite when dealing with moderate scale imagery such as MODIS. This paper directly challenges the conclusions here.

This work leverages the PANGEA projected dataset of urban land use expansion which is based on the 5 shared socioeconomic pathways (SSPs). I see no evidence for PANGEA performing any hindcast studies to estimate the uncertainties of these forecasts given that there is no reason why PANGEA could not be used to predict the past (2000-2018) and compared against the 5 yearly LULC maps. Without such a two-way analysis, it is difficult to have confidence in the results.

The LULC (Land Use Land Cover) dataset used by Guo et al. (2022) appears not to produce the same outcomes as the one employed here based on calibration with the GHSL. What are the uncertainties in these 2 datasets and could this be the reason for the divergent RF forecasts shown? What confidence can we have in these LULC datasets and what fitness for purpose has been generated for future modelling scenarios? Why is so little space devoted to mitigation measures which would be of policy impact usage such as greening-up? Are the different SSP's dealing with this aspect? The conversion of LULC from productive arable land to urban environments presupposes that policies of greening of cities will have no impact. Food can be grown hydroponically in vertical farms in cities and yet this factor is not considered.

Specific highlights of grammatical errors can be found in the uploaded annotated manuscript, more detailed comments are listed below in addition:

Note [page 5]: What is the evidence for this from empirical data?

Note [page 8]: Explain in more detail how kernels account for the effect of cloudiness?

1. Guo, T.; He, T.; Liang, S.; Roujean, J.-L.; Zhou, Y.; Huang, X. Multi-decadal analysis of high-resolution albedo changes induced by urbanization over contrasted Chinese cities based on Landsat data. Remote Sensing of Environment 2022, 269, 112832.doi:10.1016/j.rse.2021.112832

REVIEWER COMMENTS

Reviewer #3 (Remarks to the Author):

As requested by the editor, I've focused on the points raised by the first reviewer. It seems to me that the authors have carefully addressed the issues raised by the first reviewer. Remarks are taken seriously, and suggestions have been followed.

AN: Thank you for confirming that we have addressed the concerns raised by the first review. We also further elaborated the broader implications of our work (line 209-220).

Reviewer #4 (Remarks to the Author):

This paper addresses the issue of what impact will increasing urbanization have on one specific land surface parameter which will control urban heat islands, namely albedo. They employ a LULC dataset of unknown provenance and accuracy and then project forward using different GHG scenarios based on current forecasts towards 2050 and the end of the century. They do not test their model forecast using hindcasting.

AN: The model forecasting was tested and validated using hindcasting, i.e., the forecasting model was used to simulate urban expansion from 2000 to 2015, and then tested with historical urban expansion (Chen et al. 2020). The test uses FoM as an accuracy metric, which is better than the Kappa coefficient to reflect the accuracy of the land simulation. Test results show FoM values that were similar to or better than those reported by other existing land simulation applications, reflecting reliable simulation accuracy (Chen et al. 2020).

While there is no way to validate accuracy of future projections, the high-resolution projection datasets (i.e., Chen-2020) used in this study were comparable to other existing coarse resolution global urban land projections with similar spatial patterns. For example, comparing to the pattern of spatial distribution of the 0.25-degree LUH2 dataset (Hurt et al. 2006) that is officially recommended in CMIP6, it achieved a significant Pearson correlation coefficient of 0.93 (Chen et al. 2020), suggesting high spatial consistency.

This information was included in the Land Cover section in METHODS, we have further expanded the text for clarification and improvement (Lines 330-339)

By the way, The SSPs is a socio-economic scenario series that does not consider the impact of climate mitigation, although it can be coupled with the climate scenario RCPs. Therefore, the spatial impact of mitigation policies such as greening-up is not considered in Chen's projected dataset based on baseline SSPs. However, the different

SSPs have settings for whether the urban form is sprawling (e.g. SSP5) or compact (e.g. SSP1).

AN: We agree that SSPs does not directly consider the impact of climate mitigation, such as greening measures. We made it clear that the radiative forcing estimate is based on the albedo observed on the current form of existing urban land. For this reason, we have discussed possible future mitigation measures such as green or white roofs and increasing of green vegetation and blue water in new urban landscapes (Lines 248-253). Whether the urban expansion is more in the form of sprawling or in-filling (impact) is reflected in the total area of urban land under different SSP scenarios.

Recently, Guo et al. (2022 online from mid December 2021) published an analysis of 10 megacity environments within China which indicated that Radiative Forcing from these cities was POSITIVE when using fine resolution Landsat imagery over a 40 year time-scale and that these results were the opposite when dealing with moderate scale imagery such as MODIS. This paper directly challenges the conclusions here.

AN: The two studies cannot be compared because they are performed at different scales in both time and space. Firstly, our estimated positive radiative forcing was based on global lands. As shown in the maps of radiative forcing in the paper, individual regions or cities could still have negative radiative forcing. Therefore, a global positive forcing does not conflict with negative forcing found in certain cities due to albedo changes. Secondly, in our study we singled out the sole albedo change effect of converting natural land into urban land, while Guo et al. (2022)'s estimate of radiative forcing considers the whole matrix of land conversion, e.g., including the albedo effect of converting forest to cropland, which is usually a process leading to increased albedo. Lastly and most importantly, our estimate is based on annual average albedo that takes into consideration the seasonal changes of albedo, while Guo et al. (2022) is only based on summertime albedo, a time when the albedo of natural land is lowest during the year (See Figures 1&2 below as examples), which might lead to its conclusion of negative radiative forcing as urban land changes little of its albedo throughout season (See (m) from Figure 2). Note that the higher annual albedo of cropland and grassland compared to urban land is mainly because their albedo is higher than urban land during winter and fall. Therefore, using only summertime albedo difference to estimate the radiative forcing of urbanization to the climate system would be misleading. In conclusion, our results do not conflict with Guo et al. (2022). They estimated a negative radiative forcing mainly because they used summertime albedo and have included land conversions between non-urban lands. Nevertheless, we believe that in future high-resolution albedo products can reveal more spatial details and help improve estimates of climate impact from land use land cover changes.

[Redacted]

[Redacted]

This work leverages the PANGEA projected dataset of urban land use expansion which is based on the 5 shared socioeconomic pathways (SSPs). I see no evidence for PANGEA performing any hindcast studies to estimate the uncertainties of these forecasts given that there is no reason why PANGEA could not be used to predict the past (2000-2018) and compared against the 5 yearly LULC maps. Without such a two-way analysis, it is difficult to have confidence in the results.

AN: The model used to produce the projected dataset was tested and validated using hindcasting, i.e., the forecasting model was used to simulate urban expansion from 2000 to 2015 and then tested with historical urban expansion (Chen et al. 2020). The test uses FoM as an accuracy metric, which is better than the Kappa coefficient to reflect the accuracy of the land simulation. Test results show FoM values that were similar to or better than those reported by other existing land simulation applications, reflecting reliable simulation accuracy (Chen et al. 2020). The open-access version of the projected datasets has been peer-reviewed for reliability.

The LULC (Land Use Land Cover) dataset used by Guo et al. (2022) appears not to produce the same outcomes as the one employed here based on calibration with the GHSL. What are the uncertainties in these 2 datasets and could this be the reason for the divergent RF forecasts shown?

AN: The two studies cannot really be compared because they are performed at different scales in both time and space. A global positive radiative forcing from urbanization as found in our study does not exclude that at city scale; negative positive forcing exists. After carefully examine Guo et al. (2022)'s study, we think its result of negative radiative forcing may be mainly because that they only used summertime albedo observations that underestimate the year-round albedo of natural lands, especially croplands, and they also included albedo change effects between natural lands (e.g., forests to cropland, which lead to increased albedo). The use of different land-use land cover should not be the reason for the divergence, as both land use land cover products show similar land conversion propensity: urban land is mainly converted from adjacent cropland.

What confidence can we have in these LULC datasets and what fitness for purpose has been generated for future modelling scenarios?

AN: We showed high confidence in using these LULC datasets for our study purpose. Firstly, the major urbanization projection product used in the study (i.e., Chen-2020) not only was tested using historical urban expansion high agreement but also showed high spatial consistency with other urbanization projections at lower resolution, including the early projection from Seto et al. (2012) and the LUH2 dataset (Hurtt et al. 2006). Secondly, there will be spatial uncertainties in any kind of spatial simulation of land cover land use. For each year in each SSP scenario we used LULC from 100 simulations, and uncertainties have been quantified and reported in the paper (Figures 3 and 5 in the main text). Lastly, our main conclusion of a positive radiative forcing is based not only on the Chen-2020 product (i.e., the PANGEA dataset as you named) but also on two other products, Li-2017 (Li et al. 2017) and Zhou-2019 (Zhou et al. 2019). Although the magnitude of the radiative forcing is different, they all consistently suggest positive forcing. All these considerations of uncertainties enhance the confidence of our main conclusions in the study.

Why is so little space devoted to mitigation measures which would be of policy impact usage such as greening-up? Are the different SSP's dealing with this aspect? The conversion of LULC from productive arable land to urban environments presupposes those policies of greening of cities will have no impact. Food can be grown hydroponically in vertical farms in cities and yet this factor is not considered.

AN: In fact, we have substantially discussed a wide range of possible mitigation measures to reduce or even reverse the estimated positive radiative forcing caused by albedo changes in projected urbanization (see Future Mitigation Strategies section). The albedo data used in this study is based on observations from current form of urban land, which is substituted into the future and thus does not consider any changes of albedo on urban land due to possible mitigations. But if a city is already a green city, our method implies future expansion with the same green conditions. The design of SSPs may implicitly include some information about this aspect. For example, more greener cities would be expected in a more sustainable pathway. However, at least for the simulation of urban expansion, it did not and could not take this into consideration. We agree that making cities greener through vertical farms, green roofs, street trees, etc. can have multiple benefits to the urban system, biochemically and biophysically, but the extensive analysis or discussion of their impact on global climate is outside the scope of our study because there is no observation data on "future urban lands". Future studies could consider such effects related to albedo through biophysical models where the albedo of future urban land may be modelled directly from their hypothesized vertical and horizontal land compositions.

Specific highlights of grammatical errors can be found in the uploaded annotated manuscript.

AN: Thank you for finding these minor grammatical errors. We have corrected all of them. Here are the details:

1. You highlighted "magnificent" in line 59. We suppose you do not think this is a proper adjective word using here. We deleted this word and revised this sentence, which does not change the meaning of the sentence.
2. You highlighted "we choose the MODIS IGBP land cover product" in line 338. We corrected it as "we chose the MODIS IGBP land cover product".
3. You highlighted line 401 for "2018-2001" and "2030-2100". A format such as "2018-2001" where the later year is in front of the early year is used consistently to represent radiative forcing due to albedo change in the later year relative to the early year. To avoid confusion, this sentence has been rephrased as: "We analyzed the radiative forcing in 2018 due to albedo changes caused by urbanization since 2001 (2018–2001), and in the future from 2030 to 2100 at decadal intervals...."
4. Line 465, duplicated reference number was deleted.

Another comments from the annotated manuscript:

“So deforestation and afforestation are not due to human activities or urbanization? On the sentence of” Unfortunately, very limited effort has been made to study the climate effects of albedo change due to urbanization, another important LULCC process that is mainly caused by human activities”

AN: Deforestation and afforestation are, of course, caused by humans, and urbanization is part of such human activities. Our sentence says clearly that LULCC is just another (i.e., one of the) process mainly caused by human activities and does not exclude other processes due to human activities.

more detailed comments are listed below in addition:

Note [page 5]: What is the evidence for this from empirical data?

AN: This sentence states evidence based on future projections under different SSP scenarios. However, empirical data from past urban expansion also provide similar evidence for the fact that urban land mostly replaced croplands (Figure 4a) that have higher annual mean albedo than urban land (Figure S4 in supplementary). Figure S3 in supplementary suggests that the adjacency of croplands to urban lands is the main reason that urban expansion mainly replaces croplands. However, the albedo changes after urbanization vary by location, time, and original land types. To be more accurate, we changed the sentence to” These increasing trends suggest that urban expansions are projected to keep replacing lands **that on average have** higher albedo values regardless of SSP scenarios”.

Note [page 8, Line 261]: Explain in more detail how kernels account for the effect of cloudiness?

AN: The effect of cloudiness on light transmission is obvious in kernels, as cloud parameterization/representation is an essential part of General Circulation Models (Cess et al. 1996). When clouds are modeled to exist, it changes surface radiation levels, and thus the transmission which is defined as the ratio between surface radiation and top of atmosphere radiation. Our albedo kernels are derived based on climate variables simulated in General Circulation Models, and thus consider effect of cloudiness.

References used in this response:

1. Chen, G. et al. Global projections of future urban land expansion under shared socioeconomic pathways. *Nature Communications*. 11, 537 (2020).
2. Cherubini, F. et al. Spatial, seasonal, and topographical patterns of surface albedo in Norwegian forests and cropland, *International Journal of Remote Sensing*. 38, 4565-4586 (2017).
3. Guo, T. et al. Multi-decadal analysis of high-resolution albedo changes induced by urbanization over contrasted Chinese cities based on Landsat data. *Remote Sensing of Environment*. 269, 112832 (2022).
4. Hurtt, G. et al. Harmonization of global land-use change and management for the period 850--2100. <http://luh.umd.edu/> (2016).
5. Li, X. et al. A new global land-use and land-cover change product at a 1-km resolution for 2010 to 2100 based on human–environment interactions. *Annals of the American Association of Geographers*. 107 (2017)
6. Seto, K. C., Güneralp, B. & Hutyra, L. R. Global forecasts of urban expansion to 2030 and direct impacts on biodiversity and carbon pools. *Proceedings of the National Academy of Sciences of the United States of America*. 109, 16083-16088 (2012)
7. Zhang, X. et al. Analysis of global land surface shortwave broadband albedo from multiple data sources. *IEEE Journal of Selected Topics in Applied Earth Observations and Remote Sensing*. 3, 296-305 (2010).
8. Zhou, Y. et al. High-resolution global urban growth projection based on multiple applications of the SLEUTH urban growth model. *Scientific data*. 34 (2019)
9. Cess R.D. et al. Cloud feedback in atmospheric general circulation models: An update. *Journal of Geophysical Research: Atmospheres*. 101, 12791-12794 (1996)

REVIEWER COMMENTS

Reviewer #4 (Remarks to the Author):

The authors have substantially addressed all the concerns listed in the previous review no.4 for which they are thanked.

The one area of continuing disagreement concerns their argument about why Guo et al (2022) is not relevant to their global results from Chen et al (2020) and this paper. They state this is because their albedo results show the annual cycle whereas the Guo et al results show only the summer results. However, the winter results shown in Figure 1 demonstrate the effects of snow NOT the effect of changes in albedo in the urban areas.

Also, China is a touchstone for urban development given urban expansion over the last 2 decades. It is not clear that given demographic projections this will continue at the same growth rate but aside from India and Nigeria, China appears to be the country most affected.

REVIEWER COMMENTS

Reviewer #4 (Remarks to the Author):

The authors have substantially addressed all the concerns listed in the previous review no.4 for which they are thanked.

AN: Thanks for your constructive feedbacks again, which helped us improving the manuscript.

The one area of continuing disagreement concerns their argument about why Guo et al (2022) is not relevant to their global results from Chen et al (2020) and this paper. They state this is because their albedo results show the annual cycle whereas the Guo et al results show only the summer results. However, the winter results shown in Figure 1 demonstrate the effects of snow NOT the effect of changes in albedo in the urban areas.

AN: Thanks for bring up this interesting point again, which we would like to address further. We both are clear now that the main difference is that Guo et al. (2022) used summer albedo ONLY while we used annual cycle albedo. Here we would like to use this opportunity to demonstrate further that using summer albedo and annual albedo can naturally lead to divergent estimates of radiative forcing. There are two major causing factors of seasonal change of albedo: 1) the phenological changes of plants (i.e., the change of greenness/biomass/canopies), and 2) the change of weather/climate (e.g., radiation), including snow events. Snow cover is particularly critical because it can increase the albedo of snow-covered land substantially. In our last response/revision, we used Figure 1 to show the large seasonal change of albedos in natural lands and used Figure 2 to show the different effects of snow cover on vegetated lands and urban lands, which demonstrates albedo of cropland and other natural lands (e.g., forests and grassland) increased more than (and is higher than) that of urban land under snow cover conditions (therefore replacing cropland with urban land reduce albedo in winter times leading to a warming effect).

We fully agree with you on those results shown in Figure 2 (you said Figure 1, but we think you were referring to Figure 2 in our last response) demonstrating the effects of snow on albedo of different land covers. More importantly, the effect of snow is coupled with urbanization rather than isolated, as when other land covers are converted to urban land, it is changed under both snow-cover and snow-free conditions. It is therefore essential to include the effect of snow cover when computing climate effect induced by annual albedo changes. That is also the reason for us to consider the projections of snow cover under different future scenarios (RCP2.6, RCP4.5, and RCP8.5) in estimating radiative forcing for the future.

Cropland tends to have higher albedo than urban land during spring and winter due to declined greenness, change in canopy structures, and snow cover, including all

seasons or just summer can make significant differences. Here we show the global monthly average albedo of cropland vs urban land in new Figure 3, with snow cover and greenness/canopy dynamics in current real-world situation. The albedo of cropland is much higher than that of urban land during winter and spring, but the difference is the smallest during summer. The more elevated albedo at cropland than at urban land outside the summer months can lead to a warming effect when convert croplands into urban land. This overall trend can have regional differences of course but is very common in areas with distinctive seasons. In new Figure 4, we show the daily albedo changes in Beijing Metropolitan areas for a cropland and urban land pixel (MODIS 500-m resolution), respectively. A similar phenomenon of higher albedo of cropland than that of urban land in the winter is observed; while in summer times they are similar although urban land can sometimes have higher values. During snow-covered period, the albedo of cropland is much higher than that of urban land.

[Redacted]

[Redacted]

[Redacted]

Figure 3. The seasonal change of global average albedo of cropland and urban land. Data are derived based on long-term (2001-2010) averages of MODIS observations. This figure shows that in winter times, the albedo of cropland can be much higher than urban land, thus cause a warming effect of albedo when urbanizations happen. The high albedo of cropland in winter includes effects of vegetation dynamics and snow dynamics.

Figure 4. The seasonal change of MODIS pixel-wise albedo at cropland (39.679,116.221) and urban land (40.086,116.452) at local scale in Beijing Metropolitan area. Data are derived based on long-term (2015-2020) averages of MODIS observations. This figure again shows that while in summer the albedo of urban land can be higher than cropland in some time, the albedo of cropland is much higher than urban land in winter times. When both are covered by snow, the difference become larger, showing the effects of snow.

Also, China is a touchstone for urban development given urban expansion over the last 2 decades. It is not clear that given demographic projections this will continue at the same growth rate but aside from India and Nigeria, China appears to be the country most affected.

AN: We agree that China is a touchstone for urban development given urban expansion over the past 3 decades. However, it is not our focus to discuss the future trajectories of urban expansion in individual countries or continents. The growth rates of urban expansion of China have been covered in Chen et al. (2020) under different shared socioeconomic pathways, one can refer to that study if interested.

References used in this response:

1. Chen, G. et al. Global projections of future urban land expansion under shared socioeconomic pathways. *Nature Communications*. 11, 537 (2020).
2. Cherubini, F. et al. Spatial, seasonal, and topographical patterns of surface albedo in Norwegian forests and cropland, *International Journal of Remote Sensing*. 38, 4565-4586 (2017).
3. Guo, T. et al. Multi-decadal analysis of high-resolution albedo changes induced by urbanization over contrasted Chinese cities based on Landsat data. *Remote Sensing of Environment*. 269, 112832 (2022).
4. Zhang, X. et al. Analysis of global land surface shortwave broadband albedo from multiple data sources. *IEEE Journal of Selected Topics in Applied Earth Observations and Remote Sensing*. 3, 296-305 (2010).

REVIEWERS' COMMENTS

Reviewer #4 (Remarks to the Author):

Dear Authors, thank you for responding to the further questions concerning the inter-comparison of your results and those of Guo et al. (2022). I found your textual answer convincing but am concerned that you have introduced a further ambiguity. Why do you show Blue albedo rather than shortwave albedo? In your response, you show unknown albedo (blue, visible, shortwave?) in your response Fig.1 but then in Fig.2 shortwave and point to Fig. 2(i) vs 2(m) but then in your Fig.3 show Blue albedo. Is this Blue Albedo from MODIS? Is that because Blue Albedo shows the greatest effect? Also, you should consider introducing this discussion and results into your text.

REVIEWER COMMENTS

Reviewer #4 (Remarks to the Author):

Dear Authors, thank you for responding to the further questions concerning the inter-comparison of your results and those of Guo et al. (2022). I found your textual answer convincing but am concerned that you have introduced a further ambiguity. Why do you show Blue albedo rather than shortwave albedo? In your response, you show unknown albedo (blue, visible, shortwave?) in your response Fig.1 but then in Fig.2 shortwave and point to Fig. 2(i) vs 2(m) but then in your Fig.3 show Blue albedo. Is this Blue Albedo from MODIS? Is that because Blue Albedo shows the greatest effect? Also, you should consider introducing this discussion and results into your text.

AN: Thanks for pointing out the ambiguity on the use of “blue albedo” vs “shortwave albedo”. All figures in our last response were based on shortwave broadband albedo. The “blue albedo” in this context means the blue-sky shortwave albedo under the illumination of a mix of direct and diffuse light beams, in comparison with black- and white-sky radiation. We should have used the word “blue-sky albedo” more accurately to avoid this ambiguity in the previous response. “blue-sky albedo” has been consistently used in the manuscript.

In Figure 3, the albedo is derived from MODIS, which is the blue-sky shortwave albedo (i.e., not albedo of the blue wavelength band). Blue-sky albedo is composed of white-sky (from the component of diffuse beams) and black-sky albedo (from the component of direct beams). In figure 3, we plotted blue-sky albedo which is the total effects of both white-sky and black-sky albedo.

We have now included a brief discussion in our main text of seasonal changes of albedo and how it can lead to divergent results of RF estimates between using annual albedo and albedo in certain seasons.